# *NaturalSpeech 2*: Latent Diffusion Models are Natural and Zero-Shot Speech and Singing Synthesizers

**Kai Shen,**[*] **Zeqian Ju,**[*] **Xu Tan,**[*] **Yanqing Liu, Yichong Leng, Lei He**
**Tao Qin, Sheng Zhao, Jiang Bian**
Zhejiang University
Microsoft Research Asia & Microsoft Azure Speech
University of Science and Technology of China

## Abstract

Scaling text-to-speech (TTS) to large-scale, multi-speaker, and in-the-wild datasets is important to capture the diversity in human speech such as speaker identities, prosodies, and styles (e.g., singing). Current large TTS systems usually quantize speech into discrete tokens and use language models to generate these tokens one by one, which suffer from unstable prosody, word skipping/repeating issue, and poor voice quality. In this paper, we develop *NaturalSpeech 2*, a TTS system that leverages a neural audio codec with residual vector quantizers to get the quantized latent vectors and uses a diffusion model to generate these latent vectors conditioned on text input. To enhance the zero-shot capability that is important to achieve diverse speech synthesis, we design a speech prompting mechanism to facilitate in-context learning in the diffusion model and the duration/pitch predictor. We scale NaturalSpeech 2 to large-scale datasets with 44K hours of speech and singing data and evaluate its voice quality on unseen speakers. NaturalSpeech 2 outperforms previous TTS systems by a large margin in terms of prosody/timbre similarity, robustness, and voice quality in a zero-shot setting, and performs novel zero-shot singing synthesis with only a speech prompt. Audio samples are available at `https://speechresearch.github.io/naturalspeech2`.

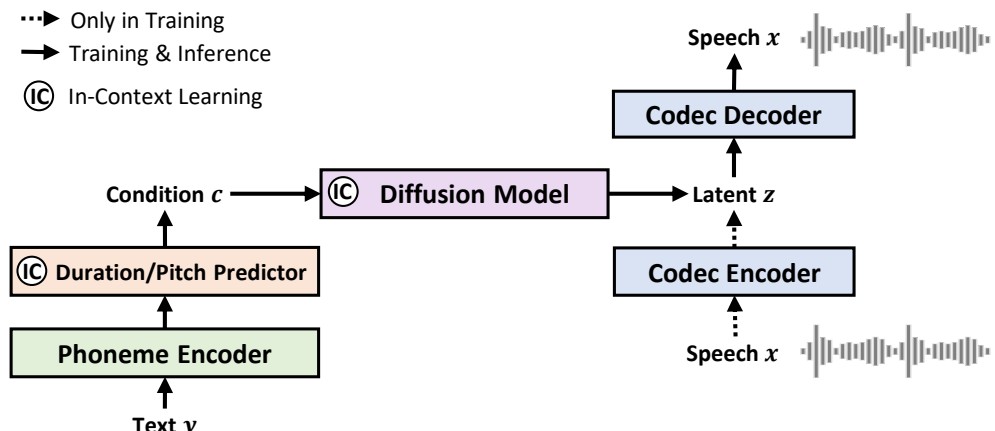

Figure 1: The overview of NaturalSpeech 2, with an audio codec encoder/decoder and a latent diffusion model conditioned on a prior (a phoneme encoder and a duration/pitch predictor). The details of in-context learning in the duration/pitch predictor and diffusion model are shown in Figure 2.

---

[*]The first three authors contributed equally to this work, and their names are listed in random order. Corresponding author: Xu Tan, `xuta@microsoft.com`

# 1 INTRODUCTION

Human speech is full of diversity, with different speaker identities (e.g., gender, accent, timbre), prosodies, styles (e.g., speaking, singing), etc. Text-to-speech (TTS) (Taylor, 2009; Tan et al., 2021) aims to synthesize natural and human-like speech with both good quality and diversity. With the development of neural networks and deep learning, TTS systems (Wang et al., 2017; Shen et al., 2018; Li et al., 2019; Ren et al., 2019; 2021a; Liu et al., 2021; 2022b; Kim et al., 2021; Tan et al., 2022) have achieved good voice quality in terms of intelligibility and naturalness, and some systems (e.g., NaturalSpeech (Tan et al., 2022; Jia et al., 2021)) even achieves human-level voice quality on single-speaker recording-studio benchmarking datasets (e.g., LJSpeech (Ito, 2017)). Given the great achievements in speech intelligibility and naturalness made by the whole TTS community, now we enter a new era of TTS where speech diversity becomes more and more important in order to synthesize natural and human-like speech.

Previous speaker-limited recording-studio datasets are not enough to capture the diverse speaker identities, prosodies, and styles in human speech due to limited data diversity. Instead, we can train TTS models on a large-scale corpus to learn these diversities, and as a by-product, these trained models can generalize to the unlimited unseen scenarios with few-shot or zero-shot technologies. Current large-scale TTS systems (Wang et al., 2023; Kharitonov et al., 2023; Xue et al., 2023) usually quantize the speech waveform into discrete tokens and model these tokens with autoregressive language models. This pipeline suffers from several limitations: 1) The speech (discrete token) sequence is usually very long (a 10s speech usually has thousands of discrete tokens) and the autoregressive models suffer from error propagation and thus unstable speech outputs. 2) There is a dilemma between the codec and language model: on the one hand, the codec with token quantization (VQ-VAE (van den Oord et al., 2017; Razavi et al., 2019) or VQ-GAN (Esser et al., 2021)) usually has a low bitrate token sequence, which, although eases the language model generation, incurs information loss on the high-frequency fine-grained acoustic details; on the other hand, some improving methods (Zeghidour et al., 2021; Défossez et al., 2022) use multiple residual discrete tokens to represent a speech frame, which increases the length of the token sequence multiple times if flattened and incurs difficulty in language modeling.

In this paper, we propose *NaturalSpeech 2*, a TTS system with latent diffusion models to achieve expressive prosody, good robustness, and most importantly strong zero-shot ability for speech synthesis. As shown in Figure 1, we first train a neural audio codec that converts a speech waveform into a sequence of latent vectors with a codec encoder, and reconstructs the speech waveform from these latent vectors with a codec decoder. After training the audio codec, we use the codec encoder to extract the latent vectors from the speech in the training set and use them as the target of the latent diffusion model, which is conditioned on prior vectors obtained from a phoneme encoder, a duration predictor, and a pitch predictor. During inference, we first generate the latent vectors from the text/phoneme sequence using the latent diffusion model and then generate the speech waveform from these latent vectors using the codec decoder.

Table 1: The comparison between NaturalSpeech 2 and previous large-scale TTS systems (Wang et al., 2023; Kharitonov et al., 2023; Xue et al., 2023).

| Methods | Previous Large-Scale Systems | NaturalSpeech 2 |
|---|---|---|
| Representations | Discrete Tokens | Continuous Vectors |
| Generative Models | Autoregressive Models | Non-Autoregressvie/Diffusion |
| In-Context Learning | Both Text and Speech are Needed | Only Speech is Needed |
| Stability/Robustness? | ✗ | ✓ |
| One Acoustic Model? | ✗ | ✓ |
| Beyond TTS (e.g., Singing)? | ✗ | ✓ |

We elaborate on some design choices in NaturalSpeech 2 (shown in Table 1) as follows.

- *Continuous vectors instead of discrete tokens.* To ensure the speech reconstruction quality of the neural codec, previous works usually quantize speech with multiple residual quantizers. As a result, the obtained discrete token sequence is very long (e.g., if using 8 residual quantizers for each speech frame, the resulting flattened token sequence will be 8 times longer), and puts much pressure on the acoustic model (autoregressive language model). Therefore, we use continuous

vectors instead of discrete tokens, which can reduce the sequence length and increase the amount of information for fine-grained speech reconstruction (see Section 3.1).

- *Diffusion models instead of autoregressive models.* We leverage diffusion models to learn the complex distributions of continuous vectors in a non-autoregressive manner and avoid error propagation in autoregressive models (see Section 3.2).

- *Speech prompting for in-context learning.* To encourage the model to follow the speech prompt characteristics and enhance the zero-shot capability, we design speech prompting mechanisms to facilitate in-context learning in the diffusion model and pitch/duration predictors (see Section 3.3).

Benefiting from these designs, NaturalSpeech 2 is more stable and robust than previous autoregressive models, and only needs one acoustic model (the diffusion model) instead of two-stage token prediction as in (Borsos et al., 2022; Wang et al., 2023), and can extend the styles beyond TTS (e.g., singing voice) due to the duration/pitch prediction and non-autoregressive generation.

We scale NaturalSpeech 2 to 400M model parameters and 44K hours of speech data, and generate speech with diverse speaker identities, prosody, and styles (e.g., singing) in zero-shot scenarios (given only a few seconds of speech prompt). Experiment results show that NaturalSpeech 2 can generate natural speech in zero-shot scenarios and outperform the previous strong TTS systems. Specifically, 1) it achieves more similar prosody with both the speech prompt and ground-truth speech; 2) it achieves comparable or better naturalness (in terms of CMOS) than the ground-truth speech on LibriSpeech and VCTK test sets; 3) it can generate singing voices in a novel timbre either with a short singing prompt, or interestingly with only a speech prompt, which unlocks the truly zero-shot singing synthesis (without a singing prompt). Audio samples can be found in `https://speechresearch.github.io/naturalspeech2`.

## 2 BACKGROUND

We present the background of NaturalSpeech 2, encompassing the pursuit of high-quality, natural voice in text-to-speech synthesis, neural audio codec models, and generative audio synthesis models.

**TTS for Natural Voice.** Text-to-speech systems (Tan et al., 2021; Wang et al., 2017; Li et al., 2019; Ren et al., 2019; Liu et al., 2021; 2022b;a; Jiang et al., 2021; Ye et al., 2023; Kim et al., 2021; Tan et al., 2022) aim to generate natural voice with both high quality and diversity. Since TTS systems have achieved good voice quality, recent works attempt to scale the TTS systems to large-scale, multi-speaker, and in-the-wild datasets to pursue diversity (Betker, 2023; Borsos et al., 2023; Zhang et al., 2023). Some works (Jiang et al., 2023b; Le et al., 2023; Li et al., 2023) propose to generate mel-spectrogram by flow-matching (Lipman et al., 2022) or GAN-based (Goodfellow et al., 2014) generation models in a NAR framework. Since the mel-spectrogram is pre-designed and intuitively less conducive to learning for neural networks, we leverage learnable latent by neural codec as the training objective. In parallel, some works (Borsos et al., 2022; Wang et al., 2023; Kharitonov et al., 2023; Xue et al., 2023; Huang et al., 2023) usually leverage a neural codec to convert speech waveform into discrete token sequence and an autoregressive language model to generate discrete tokens from text, which suffers from a dilemma that:1) Quantizing each frame into one token with vector-quantizer (VQ) (van den Oord et al., 2017; Razavi et al., 2019; Esser et al., 2021) simplifies token generation but compromises waveform quality due to high compression. 2) Quantifying each frame into multiple tokens with residual vector-quantizer (RVQ) (Zeghidour et al., 2021; Défossez et al., 2022) ensures high-fidelity waveform reconstruction but hinders autoregressive model generation due to longer token sequences, causing errors and robustness challenges. Thus, previous works, such as AudioLM (Borsos et al., 2022), leverage three-stage language models to first predict semantic tokens autoregressively, followed by generating coarse-grained tokens per frame and ultimately producing remaining fine-grained tokens. VALL-E tackles this problem using an AR model for the first codec layer tokens and an NAR model for the remaining layer tokens. These methods are complicated and incur cascaded errors. To avoid the above dilemma, we leverage a neural codec with continuous vectors and a latent diffusion model with non-autoregressive generation.

**Neural Audio Codec.** Neural audio codec (Oord et al., 2016; Valin & Skoglund, 2019; Zeghidour et al., 2021; Défossez et al., 2022) refers to a kind of neural network model that converts audio waveform into compact representations with a codec encoder and reconstructs audio waveform from these representations with a codec decoder. SoundStream (Zeghidour et al., 2021) and Encodec (Défossez

et al., 2022) leverage vector-quantized variational auto-encoders (VQ-VAE) with multiple residual vector-quantizers to compress speech into multiple tokens, and have been used as the intermediate representations for speech/audio generation (Borsos et al., 2022; Kreuk et al., 2022; Wang et al., 2023; Kharitonov et al., 2023; Xue et al., 2023). Residual vector quantizers provide good reconstruction quality and low bitrate but may not be ideal for speech/audio generation due to their long discrete token sequences ($R$ times longer if $R$ residual quantizers are used), which makes prediction tasks more challenging and may lead to errors such as word skipping, repetition, or speech collapse. In this paper, we design a neural audio codec that converts waveforms into continuous vectors, retaining fine-grained details for accurate waveform reconstruction without increasing sequence length.

**Generative Models for Speech Synthesis.** Neural TTS systems aim to synthesize high-quality voice. Generative models such as language models (Li et al., 2019; Shen et al., 2018; Wu et al., 2023), VAE (Ren et al., 2021b; Lee et al., 2022), Normalization flow (Kim et al., 2021; Miao et al., 2020; Kim et al., 2020), GAN (Kim et al., 2021; Jiang et al., 2023a), diffusion model (Kong et al., 2021; Jeong et al., 2021; Chen et al., 2021a; Popov et al., 2021; Chen et al., 2021b), self-supervised learning methods (Siuzdak et al., 2022; Du et al., 2022) and speech representation learning methods Hsu et al. (2021); Schneider et al. (2019) achieve great success. Among these, autoregressive language models and diffusion models are the two most prominent methods. Although both models are based on iterative computation (following the left-to-right process or the denoising process), autoregressive models are more sensitive to sequence length and error propagation, which cause unstable prosody and robustness issues (e.g., word skipping, repeating, and collapse). Considering text-to-speech has a strict monotonic alignment and strong source-target dependency, we leverage diffusion models enhanced with duration prediction and length expansion, which are free from robust issues.

## 3   NATURALSPEECH 2

In this section, we introduce NaturalSpeech 2, a TTS system for natural and zero-shot voice synthesis with high fidelity/expressiveness/robustness on diverse scenarios (various speaker identities, prosodies, and styles). As shown in Figure 1, NaturalSpeech 2 consists of a neural audio codec and a diffusion model with a prior model (a phoneme encoder and a duration/pitch predictor). Since speech waveform is complex and high-dimensional, following the paradigm of regeneration learning (Tan et al., 2023), we utilize post-quantized latent vectors $z$ to represent waveform $x$. Next, we employ a prior model to encode text input $y$ into a prior $c$, and a diffusion model to predict the latent vectors $z$ conditioned on prior $c$. Finally, latent vectors $z$ are input into the audio codec decoder to reconstruct the waveform $x$. We introduce the detailed designs of neural audio codec in Section 3.1 and the latent diffusion model in Section 3.2, as well as the speech prompting mechanism for in-context learning in Section 3.3.

### 3.1   NEURAL AUDIO CODEC WITH CONTINUOUS VECTORS

We use a neural audio codec to convert speech waveform into continuous vectors instead of discrete tokens, as analyzed in Section 2. Audio codec with continuous vectors enjoys several benefits: 1) Continuous vectors have a lower compression rate and higher bitrate than discrete tokens[1], which can ensure high-quality audio reconstruction. 2) Each audio frame only has one vector instead of multiple tokens as in discrete quantization, which will not increase the length of the hidden sequence.

We employ the SoundStream(Zeghidour et al., 2021) architecture as our neural audio codec, which comprises an audio encoder, a residual vector-quantizer (RVQ), and an audio decoder. The residual vector-quantizer cascades $R$ layers of vector-quantizer (VQ) and transforms the output of the audio encoder into quantized latent vectors, which serve as the training target of the diffusion model. More details about codec are provided in Appendix A.

Actually, to obtain continuous vectors, we do not need vector quantizers, but just an autoencoder or variational autoencoder. However, for regularization and efficiency purposes, we use residual vector quantizers with a very large number of quantizers and codebook tokens to approximate the continuous vectors. This provides two benefits: 1) Reduced dataset storage by storing codebook embeddings and quantized token IDs instead of high-dimensional continuous vectors, and 2) the regularization loss on discrete classification based on quantized token IDs (see $\mathcal{L}_{\text{ce-rvq}}$ in Section 3.2).

---

[1]Since our task is not compression but synthesis, we do not need a high compression rate or a low bitrate.

## 3.2 LATENT DIFFUSION MODEL WITH NON-AUTOREGRESSIVE GENERATION

We leverage a prior model, comprising a phoneme encoder, a duration predictor, and a pitch predictor, to process the text input and provide a more informative hidden vector $c$. Subsequently, the diffusion model predicts the quantized latent vector $z$ conditioned on hidden vector $c$.

**Diffusion Formulation.** We formulate the diffusion (forward) process and denoising (reverse) process (Liptser & Shiriiaev, 1977; Sohl-Dickstein et al., 2015; Ho et al., 2020) as a stochastic differential equation (SDE) (Song & Ermon, 2019; 2020; Song et al., 2020), respectively. The forward SDE transforms the latent vectors $z_0$ obtained by the neural codec (i.e., $z$) into Gaussian noises (Popov et al., 2021):

$$\mathrm{d}z_t = -\frac{1}{2}\beta_t z_t \,\mathrm{d}t + \sqrt{\beta_t}\,\mathrm{d}w_t, \quad t \in [0, 1], \tag{1}$$

where $w_t$ is the standard Brownian motion, $t \in [0, 1]$, and $\beta_t$ is a non-negative noise schedule function. Then the solution is given by:

$$z_t = e^{-\frac{1}{2}\int_0^t \beta_s ds}z_0 + \int_0^t \sqrt{\beta_s}e^{-\frac{1}{2}\int_0^t \beta_u du}\mathrm{d}w_s. \tag{2}$$

By properties of Ito's integral, the conditional distribution of $z_t$ given $z_0$ is Gaussian: $p(z_t|z_0) \sim \mathcal{N}(\rho(z_0, t), \Sigma_t)$, where $\rho(z_0, t) = e^{-\frac{1}{2}\int_0^t \beta_s ds}z_0$ and $\Sigma_t = I - e^{-\int_0^t \beta_s ds}$.

The reverse SDE transforms the Gaussian noise back to data $z_0$ with the following process:

$$\mathrm{d}z_t = -(\frac{1}{2}z_t + \nabla \log p_t(z_t))\beta_t \,\mathrm{d}t + \sqrt{\beta_t}\,\mathrm{d}\tilde{w}_t, \quad t \in [0, 1], \tag{3}$$

where $\tilde{w}$ is the reverse-time Brownian motion. Moreover, we can consider an ordinary differential equation (ODE) (Song et al., 2020) in the reverse process:

$$\mathrm{d}z_t = -\frac{1}{2}(z_t + \nabla \log p_t(z_t))\beta_t \,\mathrm{d}t, \quad t \in [0, 1]. \tag{4}$$

We can train a neural network $s_\theta$ to estimate the score $\nabla \log p_t(z_t)$ (the gradient of the log-density of noisy data), and then we can sample data $z_0$ by starting from Gaussian noise $z_1 \sim \mathcal{N}(0, 1)$ and numerically solving the SDE in Equation 3 or ODE in Equation 4. In our formulation, the neural network $s_\theta(z_t, t, c)$ is based on WaveNet (Oord et al., 2016), which takes the current noisy vector $z_t$, the time step $t$, and the condition information $c$ as input, and predicts the data $\hat{z}_0$ instead of the score, which we found results in better speech quality. Thus, $\hat{z}_0 = s_\theta(z_t, t, c)$. The loss function to train the diffusion model is as follows.

$$\mathcal{L}_{\mathrm{diff}} = \mathbb{E}_{z_0, t}[||\hat{z}_0 - z_0||_2^2 + ||\Sigma_t^{-1}(\rho(\hat{z}_0, t) - z_t) - \nabla \log p_t(z_t)||_2^2 + \lambda_{ce-rvq}\mathcal{L}_{ce-rvq}], \tag{5}$$

where the first term is the data loss, the second term is the score loss, and the predicted score is calculated by $\Sigma_t^{-1}(\rho(\hat{z}_0, t) - z_t)$, which is also used for reverse sampling based on Equation 3 or 4 in inference. The third term $\mathcal{L}_{ce-rvq}$ is a novel cross-entropy (CE) loss based on residual vector-quantizer (RVQ). Specifically, for each residual quantizer $j \in [1, R]$, we first get the residual vector $\hat{z}_0 - \sum_{i=1}^{j-1} e_i$, where $e_i$ is the ground-truth quantized embedding in the $i$-th residual quantizer. Then we calculate the L2 distance between the residual vector with each codebook embedding in quantizer $j$ and get a probability distribution with a softmax function, and then calculate the cross-entropy loss between the ID of the ground-truth quantized embedding $e_j$ and this probability distribution. $\mathcal{L}_{ce-rvq}$ is the mean of the cross-entropy loss in all $R$ residual quantizers, and $\lambda_{ce-rvq}$ is set to 0.1. Please refer to Appendix C.3 for more details of $\mathcal{L}_{ce-rvq}$.

**Prior Model: Phoneme Encoder and Duration/Pitch Predictor.** The phoneme encoder consists of 6 Transformer blocks (Vaswani et al., 2017; Ren et al., 2019), where the standard feed-forward network is modified as a convolutional network to capture the local dependency in phoneme sequence. The duration and pitch predictors utilize a similar model structure, consisting of several convolutional blocks. The ground-truth duration and pitch information is used as the learning target to train the duration and pitch predictors, with an L1 duration loss $\mathcal{L}_{\mathrm{dur}}$ and pitch loss $\mathcal{L}_{\mathrm{pitch}}$. During training, the ground-truth duration is used to expand the hidden sequence from the phoneme encoder to obtain

the frame-level hidden sequence, and then the ground-truth pitch information is added to the frame-level hidden sequence to get the final condition information $c$. During inference, the corresponding predicted duration and pitch are used.

The total loss function for the diffusion model is as follows:

$$\mathcal{L} = \mathcal{L}_{\text{diff}} + \mathcal{L}_{\text{dur}} + \mathcal{L}_{\text{pitch}}. \tag{6}$$

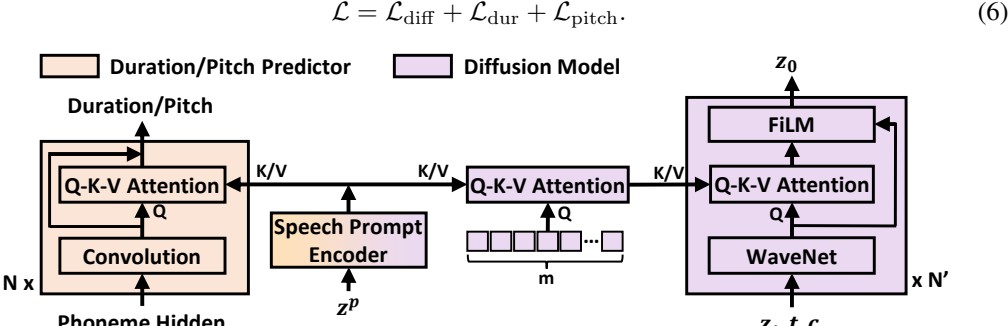

Figure 2: The speech prompting mechanism in the duration/pitch predictor and the diffusion model for in-context learning. During training, we use a random segment $z^{u:v}$ of the target speech $z$ as the speech prompt $z^p$ and use the diffusion model to only predict $z^{\backslash u:v}$. During inference, we use a reference speech of a specific speaker as the speech prompt $z^p$. Note that the prompt is the speech latent obtained by the codec encoder instead of the speech waveform.

### 3.3 Speech Prompting for In-Context Learning

To facilitate in-context learning for better zero-shot generation, we design a speech prompting mechanism to encourage the duration/pitch predictor and the diffusion model to follow the diverse information (e.g., speaker identities) in the speech prompt. For a speech latent sequence $z$, we randomly cut off a segment $z^{u:v}$ with frame index from $u$ to $v$ as the speech prompt, and concatenate the remaining speech segments $z^{1:u}$ and $z^{v:n}$ to form a new sequence $z^{\backslash u:v}$ as the learning target of the diffusion model. As shown in Figure 2, we use a Transformer-based prompt encoder to process the speech prompt $z^{u:v}$ ($z^p$ in the figure) to get a hidden sequence. To leverage this hidden sequence as the prompt, we have two different strategies for the duration/pitch predictor and the diffusion model: 1) For the duration and pitch predictors, we insert a Q-K-V attention layer in the convolution layer, where the query is the hidden sequence of the convolution layer, and the key and value is the hidden sequence from the prompt encoder. 2) For the diffusion model, instead of directly attending to the hidden sequence from the prompt encoder that exposes too many details to the diffusion model and may harm the generation, we design two attention blocks: in the first attention block, we use $m$ randomly initialized embeddings as the query sequence to attend to the prompt hidden sequence, and get a hidden sequence with a length of $m$ as the attention results (Wang et al., 2016; 2018; Yin et al., 2022); in the second attention block, we leverage the hidden sequence in the WaveNet layer as the query and the $m$-length attention results as the key and value. We use the attention results of the second attention block as the conditional information of a FiLM layer (Perez et al., 2018) to perform affine transform on the hidden sequence of the WaveNet in the diffusion model. Please refer to Appendix B for the details of WaveNet architecture used in the diffusion model.

## 4 Experiments and Results

### 4.1 Experimental Settings

In this section, we introduce experimental settings to train and evaluate NaturalSpeech 2, including the dataset, baselines, and evaluation metrics. Please refer to Appendix C for the model configuration and training and inference details.

**Datasets:** To train the neural audio codec and the diffusion model, we use the English subset of Multilingual LibriSpeech (MLS) (Pratap et al., 2020), comprising 44K hours of transcribed audiobook data. It contains 2742 male and 2748 female distinct speakers. We evaluate using two benchmark datasets: 1) LibriSpeech test-clean (Panayotov et al., 2015), with 40 distinct speakers and 5.4 hours of annotated speech; 2) VCTK dataset (Veaux et al., 2016), with 108 distinct speakers. We sample

15 and 5 utterances per speaker for LibriSpeech and VCTK, resulting in 600 and 540 evaluation utterances, respectively. For synthesis, a different same-speaker utterance is cropped into a $\sigma$-second audio segment as a prompt. All speakers in two datasets are unseen during training. Singing datasets follow a similar process, detailed in Section 4.4. See Appendix E for data processing details.

**Model Comparison:** We compare NaturalSpeech 2 with baselines including: 1) YourTTS (Casanova et al., 2022b). 2) FastSpeech 2 (Ren et al., 2021a). We adapt it by adding cross-attention on speech prompts for zero-shot synthesis. Furthermore, we also change the prediction target from the mel-spectrogram to the latent representation. 3) VALL-E (Wang et al., 2023). 4) FoundationTTS (Xue et al., 2023). 5) Voicebox (Le et al., 2023). 6) MegaTTS (Jiang et al., 2023b). For YourTTS, we use the official code and pre-trained checkpoint[2]. For FastSpeech 2, VALL-E, and FoundationTTS, we implement them according to the papers. We scale them to 400M parameters and use the same dataset for fair comparison. In addition, for Voicebox, VALL-E, and MegaTTS, since there are no official implementations, we download the audio samples from their demo page and compare them with NaturalSpeech 2 individually. Please refer to Appendix D for more details.

**Evaluation Metrics:** We use both objective and subjective metrics to evaluate the zero-shot synthesis ability of NaturalSpeech 2 and compare it with baselines. Please refer to Appendix F for a more detailed metric description.

*Objective Metrics:* 1) Prosody Similarity with Prompt. Following the practice (Zaïdi et al., 2021), we compare the difference in mean, standard deviation, skewness, and kurtosis of the duration/pitch to assess prosody similarity between synthesized and prompt speech. 2) Word Error Rate. We employ an ASR model to transcribe the generated speech and calculate the word error rate (WER).

*Subjective Metrics:* 1) Intelligibility Score. To test the robustness, following the practice in (Ren et al., 2019), we use the 50 particularly difficult sentences (see Appendix G.2) and conduct an intelligibility test. 2) CMOS and SMOS. We evaluate naturalness using comparative mean option score (CMOS), and speaker similarity using similarity mean option score (SMOS).

## 4.2 Experimental Results on Natural and Zero-Shot Synthesis

In this section, we conduct experiments comparing the NaturalSpeech 2 with the baselines in terms of: 1) *Generation Quality*, by evaluating the naturalness of the synthesized audio; 2) *Generation Similarity*, by evaluating how well the TTS system follows prompts; 3) *Robustness*, by calculating the WER and an additional intelligibility test. 4) *Generation Latency*, by evaluating the trade-off between the inference efficiency and generation quality.

**Generation Quality.** We conduct CMOS test to evaluate the generation quality (i.e., naturalness). We randomly select 20 utterances from the LibriSpeech and VCTK tests and crop the prompt speech to 3s. To ensure high-quality generation, we use a speech scoring model (Chen et al., 2022) to filter the multiple samples generated by the diffusion model with different starting Gaussian noises $z_1$. Table 2 shows a comparison of NaturalSpeech 2 against baselines and the ground truth. We have several observations: 1) NaturalSpeech 2 is comparable to the ground-truth recording in LibriSpeech ($+0.04$ is regarded as on par) and achieves much better quality on VCTK datasets ($-0.21$ is a large gap), which demonstrates the naturalness of the speech generated by NaturalSpeech 2 is high enough. 2) NaturalSpeech 2 outperforms all the baselines by a large margin in both datasets. Specifically, for VALL-E, NaturalSpeech 2 shows 0.29 and 0.31 CMOS gain in LibriSpeech and VCTK, respectively. It demonstrates that the speech generated by NaturalSpeech 2 is much more natural and of higher quality. 3) Using the cases from demo pages, we find NaturalSpeech 2 surpasses the state-of-the-art large-scale TTS systems, which shows the superiority of NaturalSpeech 2.

**Generation Similarity.** We use two metrics to evaluate the speech similarity: 1) prosody similarity between the synthesized and prompt speech. 2) SMOS test. To evaluate the prosody similarity, we randomly sample one sentence for each speaker for both LibriSpeech test-clean and VCTK dataset to form the test sets. Specifically, to synthesize each sample, we randomly and independently sample the prompt speech with $\sigma = 3$ seconds. Note that YourTTS has seen 97 speakers in VCTK in training, but we still compare NaturalSpeech 2 with YourTTS on all the speakers in VCTK (i.e., the 97 speakers are seen to YourTTS but unseen to NaturalSpeech 2).

---

[2]`https://github.com/Edresson/YourTTS`

Table 2: The CMOS, SMOS and WER results on LibriSpeech and VCTK with 95% confidence intervals. $\star$ means the results from official demo page. "-" denotes the results are not available. Note that the comparison with demo cases involves pairwise comparisons between NaturalSpeech 2 and baselines across various test cases, rendering the baseline scores in this comparison non-pairwise.

| Dataset | LibriSpeech | | | VCTK | | |
|---|---|---|---|---|---|---|
| Setting | CMOS↑ | SMOS↑ | WER↓ | CMOS↑ | SMOS↑ | WER↓ |
| Ground Truth | $+0.04$ | $4.27_{\pm 0.10}$ | 1.94 | $-0.21$ | $4.05_{\pm 0.11}$ | 9.49 |
| YourTTS (Casanova et al., 2022b) | $-0.65$ | $3.31_{\pm 0.09}$ | 7.10 | $-0.58$ | $3.39_{\pm 0.08}$ | 14.80 |
| FastSpeech 2 (Ren et al., 2021a) | $-0.53$ | $3.45_{\pm 0.08}$ | 2.10 | $-0.64$ | $3.22_{\pm 0.10}$ | 8.26 |
| FoundationTTS (Xue et al., 2023) | $-0.32$ | $3.81_{\pm 0.12}$ | 4.63 | $-0.39$ | $3.42_{\pm 0.13}$ | 12.55 |
| VALL-E (Wang et al., 2023) | $-0.29$ | $3.92_{\pm 0.11}$ | 5.72 | $-0.31$ | $3.50_{\pm 0.10}$ | 14.68 |
| NaturalSpeech 2 | **0.00** | $\mathbf{4.06}_{\pm 0.11}$ | **2.01** | **0.00** | $\mathbf{3.62}_{\pm 0.11}$ | **6.72** |
| Comparison with demo cases | | | | | | |
| VALL-E (Wang et al., 2023)$^\star$ | $-0.27$ | $3.98_{\pm 0.12}$ | - | $-0.34$ | $3.59_{\pm 0.13}$ | - |
| NaturalSpeech 2 | **0.00** | $\mathbf{4.11}_{\pm 0.11}$ | - | **0.00** | $\mathbf{3.71}_{\pm 0.12}$ | - |
| MegaTTS (Jiang et al., 2023b)$^\star$ | $-0.20$ | $3.96_{\pm 0.09}$ | - | $-0.28$ | $3.63_{\pm 0.08}$ | - |
| NaturalSpeech 2 | **0.00** | $\mathbf{4.10}_{\pm 0.11}$ | - | **0.00** | $\mathbf{3.74}_{\pm 0.10}$ | - |
| Voicebox (Le et al., 2023)$^\star$ | $-0.11$ | $3.75_{\pm 0.11}$ | - | - | - | - |
| NaturalSpeech 2 | **0.00** | $\mathbf{3.86}_{\pm 0.11}$ | - | - | - | - |

Table 3: The prosody similarity between synthesized and prompt speech in terms of the difference in mean (Mean), standard deviation (Std), skewness (Skew), and kurtosis (Kurt) of pitch and duration.

| **LibriSpeech** | Pitch | | | | Duration | | | |
|---|---|---|---|---|---|---|---|---|
| | Mean↓ | Std↓ | Skew↓ | Kurt↓ | Mean↓ | Std↓ | Skew↓ | Kurt↓ |
| YourTTS | 10.52 | 7.62 | 0.59 | 1.18 | 0.84 | **0.66** | 0.75 | 3.70 |
| FastSpeech 2 | 14.61 | 9.31 | 1.83 | 3.15 | 0.67 | 0.71 | 0.77 | 3.60 |
| FoundationTTS | 10.34 | 7.04 | 0.62 | 1.51 | 0.67 | 0.72 | 0.70 | 3.38 |
| VALL-E | 10.23 | 6.19 | 0.54 | 1.09 | **0.62** | 0.67 | 0.64 | 3.22 |
| NaturalSpeech 2 | **10.11** | **6.18** | **0.50** | **1.01** | 0.65 | 0.70 | **0.60** | **2.99** |

We apply the alignment tool to obtain phoneme-level duration and pitch and calculate the prosody similarity metrics between the synthesized speech and the prompt speech as described in Section 4.1. We report the results on LibriSpeech in Table 3 and on VCTK in Appendix H.1. We have the following observations: 1) NaturalSpeech 2 consistently outperforms all the baselines in both LibriSpeech and VCTK on most metrics, which demonstrates that our proposed NaturalSpeech 2 can mimic the prosody of prompt speech much better. 2) Although YourTTS has seen 97 from 108 speakers in VCTK dataset, our model can still outperform it by a large margin. Furthermore, we also compare prosody similarity between synthesized and ground-truth speech in Appendix H.2.

We further evaluate speaker similarity using SMOS test. We randomly select 10 utterances from LibriSpeech and VCTK datasets respectively, following the setting in the CMOS test. The prompt speech length is set to 3s. The results are shown in Table 2. We find that NaturalSpeech 2 outperforms all the baselines in two datasets. Specifically, NaturalSpeech 2 outperforms the state-of-the-art method VALL-E by 0.14 and 0.12 SMOS scores for LibriSpeech and VCTK, respectively. It demonstrates that NaturalSpeech 2 is significantly better in speaker similarity.

**Robustness.** We use the full test set of LibriSpeech and VCTK as described in Section 4.1 to synthesize the speech and compute the word error rate (WER) between the transcribed text and ground-truth text. To synthesize each sample, we use a 3-second prompt by randomly cropping the whole prompt speech. The results are shown in Table 2. We observe that: 1) NaturalSpeech 2 significantly outperforms all the baselines in LibriSpeech and VCTK, indicating better synthesis of high-quality and robust speech. 2) Our synthesized speech is comparable to the ground-truth speech in LibriSpeech and surpasses that in VCTK. The higher WER results in VCTK may stem from a noisy environment and the lack of ASR model fine-tuning in that dataset.

In addition, we conduct an intelligibility test on 50 particularly hard sentences from FastSpeech (Ren et al., 2019) to evaluate speech robustness. NaturalSpeech 2 demonstrates robustness in these cases without any intelligibility issues. Please refer to Appendix G.1 for the detailed results.

**Generation Latency.** We conduct a comparison of both the latency and generation quality with varying diffusion step ($\{20, 50, 100, 150\}$). The comparison also incorporates a NAR baseline (FastSpeech 2) and an AR model (VALL-E). As detailed in Table 12 in Appendix K, the diffusion step of 150 strikes a balance between quality (with a 0.53 CMOS gain over FastSpeech2 and a 0.29 CMOS gain over VALL-E) and latency (12.35 times faster than VALL-E). 2) As the diffusion step decreases, the inference speed increases while there is no noticeable degradation in performance. Please refer to Appendix K for more details.

### 4.3 ABLATION STUDY

In this section, we perform ablation experiments. 1) To study the effect of the speech prompt, we remove the Q-K-V attention layers in the diffusion (abbr. w/o. diff prompt), and the duration and pitch predictors (abbr. w/o. dur/pitch prompt), respectively. 2) To study the effect of the cross-entropy (CE) loss $\mathcal{L}_{ce-rvq}$ based on RVQ, we disable the CE loss by setting $\lambda_{ce-rvq}$ to 0 (abbr. w/o. CE loss). 3) To study the effectiveness of two Q-K-V attention in speech prompting for diffusion in Section 3.3, we remove the first attention that adopts $m$ randomly initialized query sequence to attend to the prompt hidden and directly use one Q-K-V attention to attend to the prompt hidden (abbr. w/o. query attn). We report CMOS and WER results in Table 4. More detailed results are in Appendix J.

We have the following observations: 1) When we disable speech prompt in diffusion, the model can not converge, which highlights its importance for high-quality TTS synthesis. 2) Disabling speech prompt in duration/pitch predictor significantly degrades audio quality (i.e., 0.45 CMOS degradation). In practice, we find that without speech prompt, it can pronounce the words correctly but with poor prosody, which causes CMOS degradation. 3) Disabling CE loss worsens both CMOS and WER performance. It shows that regularization is important for high-quality synthesis and robustness. 4) Disabling the query attention also degrades both CMOS and WER performance. In practice, we find that applying cross-attention to prompt hidden will leak details and thus mislead generation.

Table 4: The ablation study of Natural-Speech 2, measured by CMOS and WER. "-" denotes the model can not converge.

|  | CMOS | WER |
|---|---|---|
| NaturalSpeech 2 | 0.00 | 2.01 |
| w/o. diff prompt | - | - |
| w/o. dur/pitch prompt | -0.45 | 2.23 |
| w/o. CE loss | -0.25 | 3.03 |
| w/o. query attn | -0.13 | 2.65 |

### 4.4 ZERO-SHOT SINGING SYNTHESIS

In this section, we explore NaturalSpeech 2 to synthesize singing voice in a zero-shot setting, either given a singing prompt or only a speech prompt. We use speech and singing data together to train NaturalSpeech 2 with a $5e - 5$ learning rate. In inference, we set the diffusion steps to $1000$ for better performance. To synthesize a singing voice, we use the ground-truth pitch and duration, and use various singing prompts to generate singing voices with different singer timbres. Interestingly, we find that NaturalSpeech 2 can generate a novel singing voice using speech as the prompt. See the demo page[3] for zero-shot singing synthesis with either singing or speech as the prompt. Please refer to Appendix E for more details. Furthermore, we extend NaturalSpeech 2 to support more tasks such as voice conversion and speech enhancement. Please refer to Appendix L for more details.

## 5 CONCLUSION

In this paper, we develop NaturalSpeech 2, a TTS system that leverages a neural audio codec with continuous latent vectors and a latent diffusion model with non-autoregressive generation to enable natural and zero-shot text-to-speech synthesis. To facilitate in-context learning for zero-shot synthesis, we design a speech prompting mechanism in the duration/pitch predictor and the diffusion model. By scaling NaturalSpeech 2 to 400M model parameters, 44K hours of speech, and 5K speakers, it can synthesize speech with high expressiveness, robustness and strong zero-shot ability, outperforming previous TTS systems. For future work, we will explore efficient strategies such as (Song et al., 2023) to speed up, and explore large-scale speaking and singing voice training to enable more powerful mixed speaking/singing capability. We include our limitation in Appendix M.

---

[3]`https://speechresearch.github.io/naturalspeech2`

**Broader Impacts**: Since NaturalSpeech 2 could synthesize speech that maintains speaker identity, it may carry potential risks in misuse of the model, such as spoofing voice identification or impersonating a specific speaker. We conduct experiments under the assumption that the user agrees to be the target speaker in speech synthesis. If the model generalizes to unseen speakers in the real world, it should include a protocol to ensure that the speaker approves the use of their voice and a synthesized speech detection model.

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

## A  Neural Audio Codec

As shown in Figure 3, our neural audio codec consists of an audio encoder, a residual vector-quantizer (RVQ), and an audio decoder: 1) The audio encoder consists of several convolutional blocks with a total downsampling rate of 200 for 16KHz audio, i.e., each frame corresponds to a 12.5ms speech segment. 2) The residual vector-quantizer converts the output of the audio encoder into multiple residual vectors following (Zeghidour et al., 2021). The sum of these residual vectors is taken as the quantized vectors, which are used as the training target of the diffusion model. 3) The audio decoder mirrors the structure of the audio encoder, which generates the audio waveform from the quantized vectors. The working flow of the neural audio codec is as follows.

$$\text{Audio Encoder} : h = f_{\text{enc}}(x),$$

$$\text{Residual Vector Quantizer} : \{e^i_j\}^R_{j=1} = f_{\text{rvq}}(h^i), \;\; z^i = \sum_{j=1}^{R} e^i_j, \;\; z = \{z^i\}^n_{i=1}, \tag{7}$$

$$\text{Audio Decoder} : x = f_{\text{dec}}(z),$$

where $f_{\text{enc}}$, $f_{\text{rvq}}$, and $f_{\text{dec}}$ denote the audio encoder, residual vector quantizer, and audio decoder. $x$ is the speech waveform, $h$ is the hidden sequence obtained by the audio encoder with a frame length of $n$, and $z$ is the quantized vector sequence with the same length as $h$. $i$ is the index of the speech frame, $j$ is the index of the residual quantizer and $R$ is the total number of residual quantizers, and $e^i_j$ is the embedding vector of the codebook ID obtained by the $j$-th residual quantizer on the $i$-th hidden frame (i.e., $h^i$). The training of the neural codec follows the loss function in (Zeghidour et al., 2021).

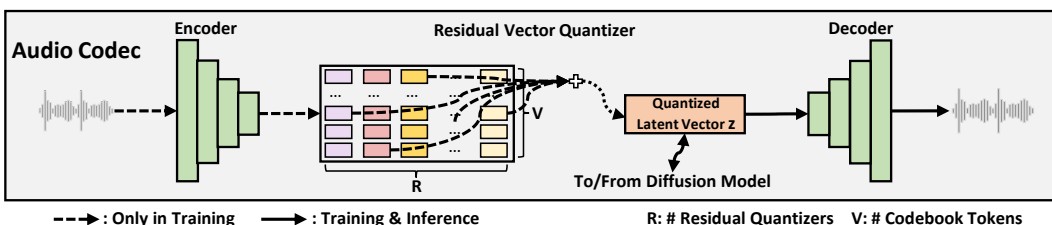

Figure 3: The neural audio codec consists of an encoder, a residual vector-quantizer (RVQ), and a decoder. The encoder extracts the frame-level speech representations from the audio waveform, the RVQ leverages multiple codebooks to quantize the frame-level representations, and the decoder takes the quantized vectors as input and reconstructs the audio waveform. The quantized vectors also serve as the training target of the latent diffusion model.

## B  The Details of WaveNet Architecture in the Diffusion Model

As shown in Figure 4, the WaveNet consists of 40 blocks. Each block consists of 1) a dilated CNN with kernel size 3 and dilation 2, 2) a Q-K-V attention, and 3) a FiLM layer. In detail, we use Q-K-V attention to attend to the key/value obtained from the first Q-K-V attention module (from the speech prompt encoder) as shown in Figure 2. Then, we use the attention results to generate the scale and bias terms, which are used as the conditional information of the FiLM layer. Finally, we average the skip output results of each layer and calculate the final WaveNet output.

## C  The Implementation Details

### C.1  Model Configuration Details

The phoneme encoder is a 6-layer Transformer (Vaswani et al., 2017) with 8 attention heads, 512 embedding dimensions, 2048 1D convolution filter size, 9 convolution 1D kernel size, and 0.1 dropout rate. The pitch and duration predictor share the same architecture of 30-layer 1D convolution with

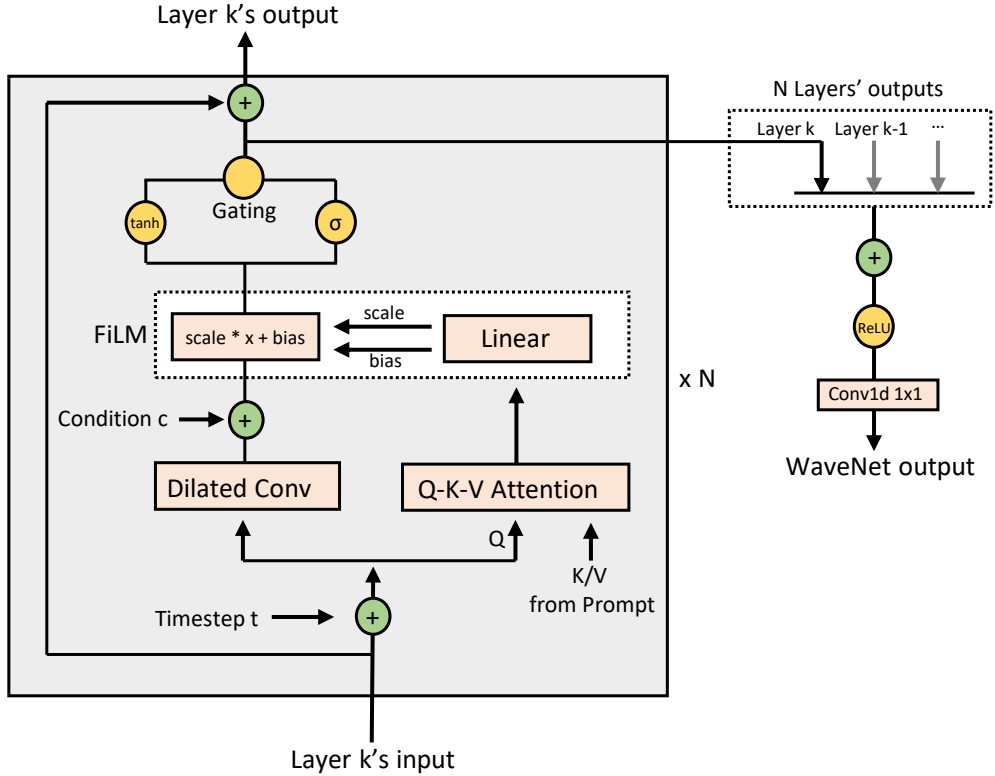

Figure 4: Overview of the WaveNet architecture in the diffusion model.

ReLU activation and layer normalization, 10 Q-K-V attention layers for in-context learning, which have 512 hidden dimensions and 8 attention heads and are placed every 3 1D convolution layers. We set the dropout to 0.5 in both duration and pitch predictors. The ground-truth pitch is quantized in the log scale and converted into pitch embedding, which is added to the expanded hidden sequence. The pitch is standardized for the learning target. For the speech prompt encoder, we use a 6-layer Transformer with 512 hidden size, which has the same architecture as the phoneme encoder. As for the $m$ query tokens in the first Q-K-V attention in the prompting mechanism in the diffusion model (as shown in Figure 2), we set the token number $m$ to 32 and the hidden dimension to 512.

The diffusion model contains 40 WaveNet layers (Oord et al., 2016), which consist of 1D dilated convolution layers with 3 kernel size, 1024 filter size, and 2 dilation size. Specifically, we use a FiLM layer (Perez et al., 2018) at every 3 WaveNet layers to fuse the condition information processed by the second Q-K-V attention in the prompting mechanism in the diffusion model. The hidden size in WaveNet is 512, and the dropout rate is 0.2.

We show the more detailed configuration in Table 5.

## C.2 MODEL TRAINING AND INFERENCE

We first train the audio codec using 8 NVIDIA TESLA V100 16GB GPUs with a batch size of 200 audios per GPU for 440K steps. We follow the implementation and experimental setting of SoundStream (Zeghidour et al., 2021) and adopt Adam optimizer with $2e - 4$ learning rate. Then we use the trained codec to extract the quantized latent vectors for each audio to train the diffusion model in NaturalSpeech 2.

The diffusion model in NaturalSpeech 2 is trained using 16 NVIDIA TESLA V100 32GB GPUs with a batch size of 6K frames of latent vectors per GPU for 300K steps. We optimize the models with the AdamW optimizer with $5e - 4$ learning rate, 32K warmup steps following the inverse square

Table 5: The detailed model configurations of NaturalSpeech 2.

| Module | Configuration | Value | #Parameters |
|---|---|---|---|
| Audio Codec | Number of Residual VQ Blocks | 16 | 27M |
| | Codebook size | 1024 | |
| | Codebook Dimension | 256 | |
| | Hop Size | 200 | |
| | Similarity Metric | L2 | |
| Phoneme Encoder | Transformer Layer | 6 | 72M |
| | Attention Heads | 8 | |
| | Hidden Size | 512 | |
| | Conv1D Filter Size | 2048 | |
| | Conv1D Kernel Size | 9 | |
| | Dropout | 0.2 | |
| Duration Predictor | Conv1D Layers | 30 | 34M |
| | Conv1D Kernel Size | 3 | |
| | Attention Layers | 10 | |
| | Attention Heads | 8 | |
| | Hidden Size | 512 | |
| | Dropout | 0.5 | |
| Pitch Predictor | Conv1D Layers | 30 | 50M |
| | Conv1D Kernel Size | 5 | |
| | Attention Layers | 10 | |
| | Attention Heads | 8 | |
| | Hidden Size | 512 | |
| | Dropout | 0.5 | |
| Speech Prompt Encoder | Transformer Layer | 6 | 69M |
| | Attention Heads | 8 | |
| | Hidden Size | 512 | |
| | Conv1D Filter Size | 2048 | |
| | Conv1D Kernel Size | 9 | |
| | Dropout | 0.2 | |
| Diffusion Model | WaveNet Layer | 40 | 183M |
| | Attention Layers | 13 | |
| | Attention Heads | 8 | |
| | Hidden Size | 512 | |
| | Query Tokens | 32 | |
| | Query Token Dimension | 512 | |
| | Dropout | 0.2 | |
| Total | | | 435M |

root learning schedule. We use a linear noise schedule function, i.e., $\beta_t = \beta_0 + (\beta_1 - \beta_0) * t$, where $\beta_0 = 0.05$ and $\beta_1 = 20$.

During inference, for the diffusion model, we find it beneficial to use a temperature $\tau$ and sample the terminal condition $z_1$ from $\mathcal{N}(0, \tau^{-1}I)$ (Popov et al., 2021). We set $\tau$ to $1.2^2$. To balance the generation quality and latency, we adopt the Euler ODE solver and set the diffusion steps to $150$. We quantize the predicted latent vectors and feed them into the audio decoder of the codec to obtain the waveform.

### C.3 THE DETAILS OF $\mathcal{L}_{\text{ce-rvq}}$

For each residual quantizer $j \in [1, R]$, we first get the residual vector:

$$z_j = z_0 - \sum_{m=1}^{j-1} \hat{e}^m, \qquad (8)$$

where $\hat{e}^m$ is the ground-truth quantized embedding in the $m$-th residual quantizer. Then we calculate the L2 distance between the residual vector with each codebook embedding in quantizer $j$ and get a probability distribution as follows:

$$l_i = ||z_j - e_i^j||_2, s_i = \frac{e^{-l_i}}{\sum_{k=1}^{N_j} e^{-l_k}}, \qquad (9)$$

where $N_j$ is the code number of residual quantizer $j$, and $s_i$ is the probability of code $i$ in codebook $j$. Finally, we can calculate the cross-entropy loss of residual quantizer $j$ given the ground-truth code index which is denoted as $L_{ce,j}$. The final CE-RVQ loss is shown as follows:

$$L_{ce-rvq} = \sum_{j=1}^{R} L_{ce,j} \qquad (10)$$

## D THE DETAILS OF BASELINE METHODS

We compare NaturalSpeech 2 with the following baselines:

- YourTTS (Casanova et al., 2022b). A powerful zero-shot TTS baseline. We use the official code and pre-trained checkpoint[4], which is trained on VCTK Veaux et al. (2016), LibriTTS Zen et al. (2019) and TTS-Portuguese Casanova et al. (2022a).

- FastSpeech 2 (Ren et al., 2021a), which is a classic high-quality TTS system. We adapt it by adding cross-attention on speech prompts for zero-shot synthesis. Furthermore, we also change the prediction target from the mel-spectrogram to the latent representation.

- FoundationTTS (Xue et al., 2023), which is another strong baseline with a neural audio codec for discrete speech token extraction and waveform reconstruction and a LLM for discrete token generation from linguistic (phoneme) tokens. To extend it to the zero-shot TTS scenario, we use an additional Transformer to encode the prompt speech features and temporally average the output to obtain a one-dimensional speaker embedding. We scale it to 400M parameters and train it on the same MLS dataset for comparison.

- VALL-E (Wang et al., 2023), which is a strong large-scale zero-shot TTS system. It uses the audio codec to discretize speech waveforms into tokens and language models to generate them. In our experiment, we implement it with reference to a third-party implementation[5]. Specifically, we use the same neural codec, dataset as used in NaturalSpeech 2 and scale it to 400 parameters for fair comparison.

- Voicebox (Le et al., 2023), which is a large-scale zero-shot TTS baseline. It uses a flow-matching (Lipman et al., 2022) model to infill speech mel-spectrogram.

- MegaTTS (Jiang et al., 2023b), which is a GAN-based large-scale zero-shot TTS system. They decompose the mel-spectrogram into different speech attributes such as timbre, and prosody, and model them according to their intrinsic properties.

Since the VALL-E, Voicebox, and MegaTTS are not open source, we download the samples from their demo pages and compare them with NaturalSpeech 2 individually. For VALL-E, we collect 8 samples in LibriSpeech and 8 samples in VCTK[6]. For Voicebox, we collect 8 samples in LibriSpeech[7]. For MegaTTS, we collect 4 samples in LibriSpeech and 4 samples in VCTK[8].

---

[4]https://github.com/Edresson/YourTTS
[5]https://github.com/lifeiteng/vall-e
[6]https://www.microsoft.com/en-us/research/project/vall-e-x/vall-e/
[7]https://voicebox.metademolab.com/zs_tts.html
[8]https://mega-tts.github.io/demo-page/

## E    The Details of Dataset

**Speech Preprocessing:**    The speech data is resampled to 16KHz. The input text sequence is first converted into a phoneme sequence using grapheme-to-phoneme conversion (Sun et al., 2019) and then aligned with speech using our internal alignment tool to obtain the phoneme-level duration. The frame-level pitch sequence is extracted from the speech using PyWorld[9].

**Singing Preprocessing:**    We collect approximately 30 hours of songs in waveform format and their corresponding lyrics from the Web, with each songs containing singing, accompaniment, backing vocals, et al.  To remove them, we employ a demucs (Défossez, 2021) model for music source separation. We apply the same duration and pitch extraction method as we apply for speech data. During the training process, we mix the singing and speech data samples. Our experimental results indicate that training the model with a mix of large-scale speech data proves to be more advantageous for enhancing its performance, as opposed to fine-tuning it on a small-scale, singing-only dataset.

## F    Evaluation Metrics

We use both objective and subjective metrics to evaluate the zero-shot synthesis ability of Natural-Speech 2 and compare it with baselines.

**Objective Metrics**    We evaluate the TTS systems with the following objective metrics:

- *Prosody Similarity with Prompt.* Following the practice (Zaïdi et al., 2021), we evaluate the prosody similarity (in terms of pitch and duration) between the generated speech and the prompt speech, which measures how well the TTS model follows the prosody in speech prompt in zero-shot synthesis. We calculate the prosody similarity with the following steps: 1) we extract phoneme-level duration and pitch from the prompt and the synthesized speech; 2) we calculate the mean, standard deviation, skewness, and kurtosis (Ren et al., 2021a) of the pitch and duration in each speech sequence; 3) we calculate the difference of the mean, standard deviation, skewness, and kurtosis between each paired prompt and synthesized speech and average the differences among the whole test set.

- *Prosody Similarity with Ground Truth.* We evaluate the prosody similarity (in terms of pitch and duration) between the generated speech and the ground-truth speech, which measures how well the TTS model matches the prosody in the ground truth. Since there is correspondence between two speech sequences, we calculate the Pearson correlation and RMSE of the pitch/duration between the generated and ground-truth speech, and average them on the whole test set.

- *Word Error Rate.* We employ an ASR model to transcribe the generated speech and calculate the word error rate (WER). The ASR model is a CTC-based HuBERT (Hsu et al., 2021) pre-trained on Librilight (Kahn et al., 2020) and fine-tuned on the 960 hours training set of LibriSpeech. We use the official code and checkpoint[10].

**Subjective Metrics**    We conduct human evaluation and use the intelligibility score and mean opinion score as the subjective metrics:

- *Intelligibility Score.* Neural TTS models often suffer from the robustness issues such as word skipping, repeating, and collapse issues, especially for autoregressive models. To demonstrate the robustness of NaturalSpeech 2, following the practice in (Ren et al., 2019), we use the 50 particularly hard sentences (see Appendix G.2) and conduct an intelligibility test. We measure the number of repeating words, skipping words, and error sentences as the intelligibility score.

- *CMOS and SMOS.* Since synthesizing natural voices is one of the main goals of NaturalSpeech 2, we measure naturalness using comparative mean option score (CMOS) with 12 native speakers as the judges. We also use the similarity mean option score (SMOS) between the synthesized and prompt speech to measure the speaker similarity, with 6 native speakers as the judges. We calculate the CMOS and SMOS by a third-party commercial evaluation platform.

---

[9]https://github.com/JeremyCCHsu/Python-Wrapper-for-World-Vocoder
[10]https://huggingface.co/facebook/hubert-large-ls960-ft

# G  INTELLIGIBILTY TEST

Table 6: The robustness of NaturalSpeech 2 and other autoregressive/non-autoregressive models on 50 particularly hard sentences. We conduct an intelligibility test on these sentences and measure the number of word repeating, word skipping, and error sentences. Each kind of word error is counted at once per sentence.

| AR/NAR | Model | Repeats | Skips | Error Sentences | Error Rate |
|---|---|---|---|---|---|
| AR | Tacotron (Wang et al., 2017) | 4 | 11 | 12 | 24% |
|  | Transformer TTS (Li et al., 2019) | 7 | 15 | 17 | 34% |
|  | VALL-E (Wang et al., 2023) | 8 | 22 | 25 | 50% |
|  | FoundationTTS (Xue et al., 2023) | 2 | 16 | 16 | 32% |
| NAR | FastSpeech (Ren et al., 2019) | 0 | 0 | 0 | 0% |
|  | NaturalSpeech (Tan et al., 2022) | 0 | 0 | 0 | 0% |
| NAR | NaturalSpeech 2 | 0 | 0 | 0 | 0% |

## G.1  INTELLIGIBILTY TEST/ROBUSTNESS TEST

Autoregressive TTS models often suffer from alignment mismatch between phoneme and speech, resulting in severe word repeating and skipping. To further evaluate the robustness of the diffusion-based TTS model, we adopt the 50 particularly hard sentences in FastSpeech (Ren et al., 2019) to evaluate the robustness of the TTS systems. We can find that the non-autoregressive models such as FastSpeech (Ren et al., 2019), NaturalSpeech (Tan et al., 2022), and also NaturalSpeech 2 are robust for the 50 hard cases, without any intelligibility issues. As a comparison, the autoregressive models such as Tacotron (Wang et al., 2017), Transformer TTS (Li et al., 2019), FoundationTTS (Xue et al., 2023), and VALL-E (Wang et al., 2023) will have a high error rate on these hard sentences. The comparison results are provided in Table 6.

## G.2  THE 50 PARTICULARLY HARD SENTENCES

The 50 particularly hard sentences used in Section G.1 are listed below:

01. a
02. b
03. c
04. H
05. I
06. J
07. K
08. L
09. 22222222 hello 22222222
10. S D S D Pass zero - zero Fail - zero to zero - zero - zero Cancelled - fifty nine to three - two - sixty four Total - fifty nine to three - two -
11. S D S D Pass - zero - zero - zero - zero Fail - zero - zero - zero - zero Cancelled - four hundred and sixteen - seventy six -
12. zero - one - one - two Cancelled - zero - zero - zero - zero Total - two hundred and eighty six - nineteen - seven -
13. forty one to five three hundred and eleven Fail - one - one to zero two Cancelled - zero - zero to zero zero Total -
14. zero zero one , MS03 - zero twenty five , MS03 - zero thirty two , MS03 - zero thirty nine ,
15. 1b204928 zero zero zero zero zero zero zero zero zero zero zero zero zero zero one seven ole32

16. zero zero zero zero zero zero zero zero two seven nine eight F three forty zero zero zero zero zero six four two eight zero one eight

17. c five eight zero three three nine a zero bf eight FALSE zero zero zero bba3add2 - c229 - 4cdb -

18. Calendaring agent failed with error code 0x80070005 while saving appointment .

19. Exit process - break ld - Load module - output ud - Unload module - ignore ser - System error - ignore ibp - Initial breakpoint -

20. Common DB connectors include the DB - nine , DB - fifteen , DB - nineteen , DB - twenty five , DB - thirty seven , and DB - fifty connectors .

21. To deliver interfaces that are significantly better suited to create and process RFC eight twenty one , RFC eight twenty two , RFC nine seventy seven , and MIME content .

22. int1 , int2 , int3 , int4 , int5 , int6 , int7 , int8 , int9 ,

23. seven _ ctl00 ctl04 ctl01 ctl00 ctl00

24. Http0XX , Http1XX , Http2XX , Http3XX ,

25. config file must contain A , B , C , D , E , F , and G .

26. mondo - debug mondo - ship motif - debug motif - ship sts - debug sts - ship Comparing local files to checkpoint files ...

27. Rusbvts . dll Dsaccessbvts . dll Exchmembvt . dll Draino . dll Im trying to deploy a new topology , and I keep getting this error .

28. You can call me directly at four two five seven zero three seven three four four or my cell four two five four four four seven four seven four or send me a meeting request with all the appropriate information .

29. Failed zero point zero zero percent < one zero zero one zero zero zero zero Internal . Exchange . ContentFilter . BVT ContentFilter . BVT_log . xml Error ! Filename not specified .

30. C colon backslash o one two f c p a r t y backslash d e v one two backslash oasys backslash legacy backslash web backslash HELP

31. src backslash mapi backslash t n e f d e c dot c dot o l d backslash backslash m o z a r t f one backslash e x five

32. copy backslash backslash j o h n f a n four backslash scratch backslash M i c r o s o f t dot S h a r e P o i n t dot

33. Take a look at h t t p colon slash slash w w w dot granite dot a b dot c a slash access slash email dot

34. backslash bin backslash premium backslash forms backslash r e g i o n a l o p t i o n s dot a s p x dot c s Raj , DJ ,

35. Anuraag backslash backslash r a d u r five backslash d e b u g dot one eight zero nine underscore P R two h dot s t s contains

36. p l a t f o r m right bracket backslash left bracket f l a v o r right bracket backslash s e t u p dot e x e

37. backslash x eight six backslash Ship backslash zero backslash A d d r e s s B o o k dot C o n t a c t s A d d r e s

38. Mine is here backslash backslash g a b e h a l l hyphen m o t h r a backslash S v r underscore O f f i c e s v r

39. h t t p colon slash slash teams slash sites slash T A G slash default dot aspx As always , any feedback , comments ,

40. two thousand and five h t t p colon slash slash news dot com dot com slash i slash n e slash f d slash two zero zero three slash f d

41. backslash i n t e r n a l dot e x c h a n g e dot m a n a g e m e n t dot s y s t e m m a n a g e

42. I think Rich's post highlights that we could have been more strategic about how the sum total of XBOX three hundred and sixtys were distributed .

43. 64X64 , 8K , one hundred and eighty four ASSEMBLY , DIGITAL VIDEO DISK DRIVE , INTERNAL , 8X ,

44. So we are back to Extended MAPI and C++ because . Extended MAPI does not have a dual interface VB or VB .Net can read .

45. Thanks , Borge Trongmo Hi gurus , Could you help us E2K ASP guys with the following issue ?

Table 7: The prosody similarity between synthesized and prompt speech in terms of the difference in mean (Mean), standard deviation (Std), skewness (Skew), and kurtosis (Kurt) of pitch and duration on VCTK.

| VCTK | Pitch | | | | Duration | | | |
|---|---|---|---|---|---|---|---|---|
| | Mean↓ | Std↓ | Skew↓ | Kurt↓ | Mean↓ | Std↓ | Skew↓ | Kurt↓ |
| YourTTS | 13.67 | 6.63 | 0.72 | 1.54 | **0.72** | 0.85 | 0.84 | 3.31 |
| FastSpeech 2 | 18.17 | 9.87 | 2.04 | 3.67 | 0.81 | 0.79 | 0.86 | 3.12 |
| FoundationTTS | 13.41 | 6.59 | 0.76 | 1.68 | 0.80 | 0.82 | 0.80 | 3.38 |
| VALL-E | 13.33 | 6.44 | 0.73 | 1.36 | 0.74 | 0.79 | 0.82 | 2.91 |
| NaturalSpeech 2 | **13.29** | **6.41** | **0.68** | **1.27** | 0.79 | **0.76** | **0.76** | **2.65** |

46. Thanks J RGR Are you using the LDDM driver for this system or the in the build XDDM driver ?

47. Btw , you might remember me from our discussion about OWA automation and OWA readiness day a year ago .

48. empidtool . exe creates HKEY_CURRENT_USER Software Microsoft Office Common QMPersNum in the registry , queries AD , and the populate the registry with MS employment ID if available else an error code is logged .

49. Thursday, via a joint press release and Microsoft AI Blog, we will announce Microsoft's continued partnership with Shell leveraging cloud, AI, and collaboration technology to drive industry innovation and transformation.

50. Actress Fan Bingbing attends the screening of 'Ash Is Purest White (Jiang Hu Er Nv)' during the 71st annual Cannes Film Festival

# H  PROSODY SIMILARITY RESULTS

## H.1  PROSODY SIMILARITY WITH PROMPT SPEECH ON VCTK

In this section, we report the prosody similarity evaluation results on VCTK in Table 7.

## H.2  PROSODY SIMILARITY WITH GROUND TRUTH

To further investigate the quality of prosody, we follow the generation quality evaluation of *prosody similarity between synthesized and prompt speech* in Section 4.2 and compare the generated speech with the ground-truth speech. We use Pearson correlation and RMSE to measure the prosody matching between generated and ground-truth speech. The results are shown in Table 8. We observe that NaturalSpeech 2 outperforms all baselines by a large margin, which shows that our NaturalSpeech 2 is much better in prosody similarity.

# I  EXPERIMENTS ON PROMPT LENGTH

Since the prompt length is an important hyper-parameter for zero-shot TTS, we would like to investigate the effect of the prompt length. We follow the setting of *prosody similarity between synthesized and prompt speech* in Section 4.2. Specifically, we vary the prompt length by $\sigma = \{3, 5, 10\}$ seconds and report the prosody similarity metrics of NaturalSpeech 2. The results are shown in Table 9. We observe that when the prompt is longer, the similarity between the generated speech and the prompt is higher for NaturalSpeech 2. It shows that the longer prompt reveals more details of the prosody, which help the TTS model to generate more similar speech.

# J  ABLATION STUDY

In this section, we present the detailed ablation results by evaluating the prosody similarity between synthesized audio generated by the ablation model and the prompt speech, which are conducted in Section 4.3. The results are shown in Table 10.

Table 8: The prosody similarity between the synthesized and ground-truth speech in terms of the correlation and RMSE on pitch and duration.

| LibriSpeech | Pitch | | Duration | |
|---|---|---|---|---|
| | Correlation ↑ | RMSE ↓ | Correlation ↑ | RMSE ↓ |
| YourTTS | 0.77 | 51.78 | 0.52 | 3.24 |
| FastSpeech 2 | 0.64 | 60.39 | 0.63 | 2.92 |
| FoundationTTS | 0.73 | 52.18 | 0.61 | 3.16 |
| VALL-E | 0.73 | 50.80 | 0.62 | 2.88 |
| NaturalSpeech 2 | **0.81** | **47.72** | **0.65** | **2.72** |
| VCTK | Pitch | | Duration | |
| | Correlation ↑ | RMSE ↓ | Correlation ↑ | RMSE ↓ |
| YourTTS | 0.82 | 42.63 | 0.55 | 2.55 |
| FastSpeech 2 | 0.77 | 47.40 | 0.60 | 2.63 |
| FoundationTTS | 0.81 | 46.00 | 0.53 | 2.64 |
| VALL-E | 0.83 | 43.27 | 0.61 | 2.52 |
| NaturalSpeech 2 | **0.87** | **39.83** | **0.64** | **2.50** |

Table 9: The NaturalSpeech 2 prosody similarity between the synthesized and prompt speech with different lengths in terms of the difference in the mean (Mean), standard deviation (Std), skewness (Skew), and kurtosis (Kurt) of pitch and duration.

| LibriSpeech | Pitch | | | | Duration | | | |
|---|---|---|---|---|---|---|---|---|
| | Mean↓ | Std↓ | Skew↓ | Kurt↓ | Mean↓ | Std↓ | Skew↓ | Kurt↓ |
| 3s | 10.11 | 6.18 | 0.50 | 1.01 | 0.65 | 0.70 | 0.60 | 2.99 |
| 5s | 6.96 | 4.29 | 0.42 | 0.77 | 0.69 | 0.60 | 0.53 | 2.52 |
| 10s | 6.90 | 4.03 | 0.48 | 1.36 | 0.62 | 0.45 | 0.56 | 2.48 |
| VCTK | Pitch | | | | Duration | | | |
| | Mean↓ | Std↓ | Skew↓ | Kurt↓ | Mean↓ | Std↓ | Skew↓ | Kurt↓ |
| 3s | 13.29 | 6.41 | 0.68 | 1.27 | 0.79 | 0.76 | 0.76 | 2.65 |
| 5s | 14.46 | 5.47 | 0.63 | 1.23 | 0.62 | 0.67 | 0.74 | 3.40 |
| 10s | 10.28 | 4.31 | 0.41 | 0.87 | 0.71 | 0.62 | 0.76 | 3.48 |

Furthermore, we also compare the prosody similarity between audio generated by the ablation model and the ground-truth speech in Table 11. Similar to the results of comparing the audio generated by the ablation model and prompt speech, we also have the following observations. 1) The speech prompt is most important to the generation quality. 2) The cross-entropy and the query attention strategy are also helpful in high-quality speech synthesis.

## K    LATENCY STUDY OF NATURALSPEECH 2

In this section, we report the inference latency of NaturalSpeech 2. We vary the diffusion step in $\{20, 50, 100, 150\}$, and report both the latency (RTF) and generation quality (CMOS). We also compare NaturalSpeech 2 with a NAR baseline (FastSpeech 2) and an AR model (VALL-E). The latency tests are conducted on a server with E5-2690 Intel Xeon CPU, 512GB memory, and one NVIDIA V100 GPU. The results are shown in Table 12.

From the results, we have several observations. 1) When the diffusion step is 150 (used in our paper), NaturalSpeech 2 is 33.3 times slower than the NAR model FastSpeech 2, but achieves 0.53 CMOS gain. Still, it is 12.35 times faster than VALL-E. 2) NaturalSpeech 2 has 0.08 CMOS drop and

Table 10: The ablation study of NaturalSpeech 2. The prosody similarity between the synthesized and prompt speech in terms of the difference in the mean (Mean), standard variation (Std), skewness (Skew), and kurtosis (Kurt) of pitch and duration. "-" denotes the model can not converge.

| | Pitch | | | | Duration | | | |
|---|---|---|---|---|---|---|---|---|
| | Mean↓ | Std↓ | Skew↓ | Kurt↓ | Mean↓ | Std↓ | Skew↓ | Kurt↓ |
| NaturalSpeech 2 | **10.11** | **6.18** | **0.50** | **1.01** | **0.65** | **0.70** | **0.60** | **2.99** |
| w/o. diff prompt | - | - | - | - | - | - | - | - |
| w/o. dur/pitch prompt | 21.69 | 19.38 | 0.63 | 1.29 | 0.77 | 0.72 | 0.70 | 3.70 |
| w/o. CE loss | 10.69 | 6.24 | 0.55 | 1.06 | 0.71 | 0.72 | 0.74 | 3.85 |
| w/o. query attn | 10.78 | 6.29 | 0.62 | 1.37 | 0.67 | 0.71 | 0.69 | 3.59 |

Table 11: The ablation study of NaturalSpeech 2. The prosody similarity between the synthesized and ground-truth speech in terms of the correlation and RMSE on pitch and duration. "-" denotes that the model can not converge.

| | Pitch | | Duration | |
|---|---|---|---|---|
| | Correlation ↑ | RMSE ↓ | Correlation ↑ | RMSE ↓ |
| NaturalSpeech 2 | **0.81** | **47.72** | **0.65** | **2.72** |
| w/o. diff prompt | - | - | - | - |
| w/o. dur/pitch prompt | 0.80 | 55.00 | 0.59 | 2.76 |
| w/o. CE loss | 0.79 | 50.69 | 0.63 | 2.73 |
| w/o. query attn | 0.79 | 50.65 | 0.63 | 2.73 |

2.95 times speedup (compared with 150 steps) when the diffusion step is 50. The CMOS drops 0.21 while it can achieve 7.31 times speedup (compared with 150 steps) when the diffusion step is 20. Furthermore, since NaturalSpeech 2 is parallel to many diffusion speedup works such as the consistency model Song et al. (2023), we will explore speeding up the diffusion model while retaining the generation quality in the future.

## L  VOICE CONVERSION AND SPEECH ENHANCEMENT

### L.1  VOICE CONVERSION

Besides zero-shot text-to-speech and singing synthesis, NaturalSpeech 2 also supports zero-shot voice conversion, which aims to convert the source audio $z_{source}$ into the target audio $z_{target}$ using the voice of the prompt audio $z_{prompt}$. Technically, we first convert the source audio $z_{source}$ into an informative Gaussian noise $z_1$ using a *source-aware diffusion process* and generate the target audio $z_{target}$ using a *target-aware denoising process*, shown as follows.

**Source-Aware Diffusion Process**  In voice conversion, it is helpful to provide some necessary information from source audio for target audio in order to ease the generation process. Thus, instead of directly diffusing the source audio with some Gaussian noise, we diffuse the source audio into a starting point that still maintains some information in the source audio. Specifically, inspired by the stochastic encoding process in Diffusion Autoencoder (Preechakul et al., 2022), we obtain the starting point $z_1$ from $z_{source}$ as follows:

$$z_1 = z_0 + \int_0^1 -\frac{1}{2}(z_t + \Sigma_t^{-1}(\rho(\hat{s}_\theta(z_t, t, c), t) - z_t))\beta_t \, dt, \tag{11}$$

where $\Sigma_t^{-1}(\rho(\hat{s}_\theta(z_t, t, c), t) - z_t)$ is the predicted score at $t$. We can think of this process as the reverse of ODE (Equation 4) in the denoising process.

Table 12: The latency study of NaturalSpeech 2. We report the RTF and CMOS results for different diffusion steps.

| Model | Diffusion Step | RTF | CMOS |
|---|---|---|---|
| NaturalSpeech 2 | 150 | 0.366 | 0.00 |
| NaturalSpeech 2 | 100 | 0.244 | -0.02 |
| NaturalSpeech 2 | 50 | 0.124 | -0.08 |
| NaturalSpeech 2 | 20 | 0.050 | -0.21 |
| FastSpeech 2 | - | 0.011 | -0.53 |
| VALL-E | - | 4.52 | -0.29 |

**Target-Aware Denoising Process** Different from the TTS which starts from the random Gaussian noise, the denoising process of voice conversion starts from the $z_1$ obtained from the source-aware diffusion process. We run the standard denoising process as in the TTS setting to obtain the final target audio $z_{target}$, conditioned on $c$ and the prompt audio $z_{prompt}$, where $c$ is obtained from the phoneme and the duration sequence of the source audio and the predicted pitch sequence.

### L.2 SPEECH ENHANCEMENT

NaturalSpeech 2 can be extended to speech enhancement, which is similar to the extension of voice conversion. In this setting, we assume that we have the source audio $z'_{source}$ which contains background noise ( $z'$ denotes the audio with background noise), the prompt with background noise $z'_{prompt}$ for *the source-aware diffusion process*, and the prompt without background noise $z_{prompt}$ for *target-aware denoising process*. Note that $z'_{source}$ and $z'_{prompt}$ have the same background noise.

To remove the background noise, firstly, we apply the *source-aware diffusion process* by $z'_{source}$ and $z'_{prompt}$ and obtain the $z_1$ as in Equation 11. The source audio's duration and pitch are utilized in this procedure. Secondly, we run the *target-aware denoising process* to obtain the clean audio by $z_1$ and the clean prompt $z_{prompt}$. Specifically, we use the phoneme sequence, duration sequence, and pitch sequence of the source audio in this procedure.

## M    LIMITATION AND FUTURE WORKS

Despite NaturalSpeech 2 has made great progress, it still suffers from the following issues.

**Data coverage.** Although the 44K speech data from MLS dataset is large compared to previous works, they can not cover everyone's voice. In audiobooks, most speakers will read the books clearly and fluently, while in real world people will speak causally, thus leading to degradation when generalizing to real-world scenarios. In the future, we will scale NaturalSpeech 2 to more generalized and larger-scale benchmarks to enhance the zero-shot generation ability.

**Inference efficiency.** Although NaturalSpeech 2 is a non-autoregressive generation model, it still needs multiple iterations during inference. In the future, we will explore efficient strategies such as consistency models to speed up the diffusion model.

**Singing voice quality.** Although NaturalSpeech 2 can synthesize singing voices in a zero-shot manner, the quality is not as good as the TTS synthesis's quality. We think we are limited in two aspects: 1) the scale of singing data and 2) the quality of singing data. For data scale, we only collect 30 hours, which is small compared with the speech data scale. For data quality, it is difficult to obtain clean human voices from commercial songs, which are a combination of vocals, backing vocals, accompaniment, and other background noises. In the future, we will first explore more efficient methods to collect more singing data and obtain higher-quality singing voices to enhance the singing voice quality.

