# OpenReview forum: "NaturalSpeech 2: Latent Diffusion Models are Natural and Zero-Shot Speech and Singing Synthesizers"
_ICLR.cc/2024/Conference — ICLR 2024 spotlight_

### Official Review · Reviewer_8Wev · 2023-10-30

**Soundness:** 4 excellent
**Presentation:** 4 excellent
**Contribution:** 3 good
**Rating:** 8
**Confidence:** 5

**Summary:**

This work proposes a latent diffusion-based speech synthesis framework for high-quality zero-shot speech synthesis. They utilize an audio codec as a latent representation and a conditional latent diffusion model could generate a latent representation. Then, the codec decoder generates a waveform audio. The zero-shot results show a better performance than the codec-based TTS model and YourTTS. Moreover, the audio quality is good.

**Strengths:**

They propose the latent diffusion-based speech synthesis model. This work may be the first successful implementation of a latent diffusion model for speech synthesis. Although recently large-language model (LLM) -based speech synthesis models have been investigated, they have too many problems for speech synthesis resulting from the auto-regressive generative manner. However, this work adopts a parallel synthesis framework with latent diffusion, and successfully shows their generative performance by several speech tasks.

Recent papers only compare their work with YourTTS but I do not think YourTTS is a good zero-shot TTS model. The audio quality of YourTTS is too bad. However, although recent models do not provide an official implementation, the authors tried to compare their model with many other works.

**Weaknesses:**

1. They also conducted an ablation study well. However, it would be better if the authors could add the results according to the dataset and model size. The model size of NaturalSpeech 2 is too complex compared to VITS. In my personal experience, VITS with speaker prompt could achieve significantly better performance than YourTTS.

2. For inference speed, NaturalSpeech 2 still has a low latency for its iterative generation. Although this discussion is included in the Appendix, it would be better if the authors could add the discussion of inference speed in the main text. This is just a limitation of diffusion models so I acknowledge the trade-off between quality and inference speed. Furthermore, I hope to know other metrics of NaturaSpeech 2 according to Diffusion Steps (WER or Similarity metric). Recently, Flow matching using optimal transport is utilized for fast speech synthesis. This could be adopted to this work.

3. Some details are missing. Please see the questions.

**Questions:**

1. This work utilizes a quantized latent vector for latent representation. In my experience, the quality of the model with the continuous latent representation before quantization showed a better performance in latent diffusion model for singing voice synthesis. Have you tried to train your model with the pre- or post-quantized representation for latent representation?

2. The details of singing voice synthesis are missing. It would be better if you could add the details for pre-processing of musical scores. How do you extract the duration of phonemes in this work?

3. How do you extract the pitch information? This significantly affects the performance so the details should be included. (about F0 min, F0 max, resolution, and pitch extraction algorithm).

4. The authors may train the audio codec with their speech dataset. I think it is important to utilize a high-quality speech codec for high-quality speech synthesis. In this regard, I hope that the authors will mention about this part by comparing your model with the same model utilizing an official Soundstream codec as a latent representation.

---

> ### Author Response · Authors · 2023-11-19
> **Response to Reviewer 8Wev**
>
> First of all, we want to thank the reviewer for your careful reading and providing a lot of constructive comments! Below we address the concerns mentioned in the review.
>
> $\textbf{Q1}$: It would be better if the authors could add the results according to the dataset and model size.
>
> $\textbf{A1}$: This is a great suggestion. We conduct serial experiments by varying model sizes and data sizes. The results are shown as follows:
> | Model Size | Data Size | CMOS |
> | -------------- | ----- | ----- |
> | 400M | 44,000 h | 0.00 |
> | 400M | 10,000 h | -0.06 |
> | 400M | 1,000 h | -0.41 |
> | 60M | 1,000 h | -0.52 |
>
> We have the following observations.
> * When we decrease the data size from 44k hours to 1k hours while keeping the model size unchanged, we can find the CMOS drops consistently. It indicates that the data size is important to generate high-quality speech.
> * When we further decrease the model size from 400M to 60M while keeping the data size unchanged, the CMOS drops 0.11, which indicates the model size is important as well.
>
> $\textbf{Q2}$: Add discussion of inference speed in the main text. Furthermore, I hope to know other metrics of NaturaSpeech 2 according to Diffusion Steps (WER or Similarity metric).  And suggestions for flow matching.
>
> $\textbf{A2}$: Thank you for your valuable suggestion.  Firstly, we incorporate this discussion in the revised version as suggested.
>
> Secondly, for additional metrics, we report SMOS and WER results in the table below. These results are in line with CMOS, further illustrating the trade-off between quality and inference speed.
> | Diffusion Step | RTF   | CMOS  | SMOS  | WER  |
> | -------------- | ----- | ----- | --------------- | ---- |
> | 150            | 0.366 | 0.00  | 4.06 | 2.01 |
> | 100            | 0.244 | -0.02 | 4.04 | 2.04 |
> | 50             | 0.124 | -0.08 | 3.98 | 2.09 |
> | 20             | 0.050 | -0.21 | 3.88 | 2.15 |
>
> Finally, as suggested, we would like to explore efficient strategies such as the consistency model[1] and flow matching model[2] to speed up the diffusion model.
>
> [1] Song, Yang, et al. Consistency models.
>
> [2] Lipman, Yaron, et al. Flow matching for generative modeling.
>
> $\textbf{Q3}$: This work utilizes a quantized latent vector for latent representation. Have you tried to train your model with the pre- or post-quantized representation for latent representation?
>
> $\textbf{A3}$: Thanks for your careful reading. In the preliminary experiment, we did not observe performance improvements when replacing the post-quantized representation with the pre-quantized representation.
>
> Moreover, using post-quantized representation is driven by engineering considerations. For example, rather than directly storing 256-dimensional FP32 (256 x 32 bits) latent representations, we only needed to store 16 INT16 quantized IDs (16 x 16 bits), leading to a 32x boost in storage efficiency.
>
> $\textbf{Q4}$:  How do you extract the duration of phonemes in this work?
>
> $\textbf{A4}$: Thanks for your query. We would like to clarify that our singing voice data comes in the form of recorded audio, not musical scores. We use the same alignment tool for extracting duration information from singing voice data. We have updated the paper and please refer to the revision in Appendix E.
>
> $\textbf{Q5}$: How do you extract the pitch information?
>
> $\textbf{A5}$: We appreciate your question. The pitch of each frame is extracted using the pyworld[1] with the dio + stonemask algorithm. The parameters are set to default.
>
> [1] https://github.com/JeremyCCHsu/Python-Wrapper-for-World-Vocoder
>
> $\textbf{Q6}$: The authors may train the audio codec with their speech dataset. I hope that the authors will mention about this part by comparing your model with the same model utilizing an official SoundStream codec as a latent representation.
>
> $\textbf{A6}$: We appreciate your thorough understanding and insightful comments. Since we could not find the official pytorch implementation of SoundStream,  we attempted to reproduce it and the evaluation results are shown as follows (in the 12.8kbps setting):
> | Implementation | VISQOL |
> | --- | --- |
> | Ours                  |  4.38      |
> | SoundStream Paper | 4.26 |
>
> From the results, we can find that our implementation is competitive with the original performance.

---

> ### Comment · Reviewer_8Wev · 2023-11-20
> **Thanks for your response**
>
> Thanks for addressing my concern. I will keep my original score of 8.
>
> However, I have a question about the term of singing voice synthesis. I hope to confirm the definition of singing voice synthesis. The goal of singing voice synthesis is synthesizing the singing from musical note, not from the recorded singing voice isn't it right? If it is, the term of singing synthesis is overclaimed. The task in Section 4.4 is just singing voice conversion which is relatively easy task. The term may not be defined yet so I just hope to know your opinion.

---

> ### Author Response · Authors · 2023-11-20
> **Response to Reviewer 8Wev**
>
> Thanks for the question. We think there may be some misunderstanding. Firstly, we do not generate the singing from the recording, but from the lyric, ground truth duration, and ground truth pitch. Thus it is not singing voice conversion. Secondly, in practice, the ground truth pitch/duration can be determined by a separately-trained neural network conditioned on musical scores. Following [1,2], we use the ground truth pitch/duration during inference for simplification.
>
> [1] Liu, Jinglin, et al. Diffsinger: Singing voice synthesis via shallow diffusion mechanism.
>
> [2] Huang, Rongjie, et al. Make-A-Voice: Unified Voice Synthesis With Discrete Representation.

---

> ### Comment · Reviewer_8Wev · 2023-11-20
> **Thanks for your response**
>
> Thanks for your response.
>
> I acknowledged that the authors utilized a duration and pitch to synthesize a singing voice, not from the recorded audio. However, they may be extracted from the recorded audio; therefore, they require the ground-truth singing voice to generate a singing voice. I know that this is a simple way to generate a singing voice by ground truth frame-level duration and pitch from the ground-truth audio, however what if I hope to generate the singing voice without ground-truth audio. In this regard, I stated it as a singing voice conversion which requires a ground-truth audio to extract a ground-truth duration and pitch as input representations.
>
> I just hope to discuss it. I'm not talking whether it is right or wrong! Feel free to response this :)

---

> > ### Author Response · Authors · 2023-11-20
> >
> > Thanks for your response.
> > We agree with your opinion. Strictly speaking, defining SVS in this way is not rigorous, as we use ground truth audio to obtain duration/pitch. The duration/pitch should be determined by a separately-trained neural network conditioned on musical scores instead of directly obtained from singing audio. However, in NaturalSpeech 2, it is not our main focus, thus we directly use the duration/pitch from the audio. Inspired by this discussion, we would like to explore it in the future.

---

> ### Author Response · Authors · 2023-11-22
>
> We would like to further express our sincere thanks for your insightful suggestion and engaging discussion! We truly value your time and efforts in the reviewing process.

---

### Official Review · Reviewer_69Md · 2023-10-31

**Soundness:** 4 excellent
**Presentation:** 3 good
**Contribution:** 3 good
**Rating:** 8
**Confidence:** 4

**Summary:**

The paper proposes a new TTS model that is capable of generating speech with diverse speaker identities, prosody, and styles, in zero-shot scenarios and it can also sing. It outperforms the current SOTA methods in both objective and subjective metrics. The way it works is the following. First the neural audio codec that converts a speech waveform into a sequence of latent vectors with a codec encoder, and reconstructs the speech waveform from these latent vectors with a codec decoder. Then the codec encoder extracts the latent vectors from the speech and uses them as the target of the latent diffusion model which is conditioned on prior vectors. During inference it generates the latent vectors from text using the diffusion model and then generate the speech waveform using the codec decoder.

**Strengths:**

-Paper is very well written and provides good intuition and justification for all model choices that the authors have made. These choices are intuitive to make generated speech more natural and to overcome past bottlenecks in previous methods.
-The new TTS algorithm has many capabilities such as generating diverse speech (different speakers, prosody, style) and in zero-shot scenarios. Singing is a bonus in this case.
-NaturalSpeech2 beats current SOTA methods in both objective and subjective metrics.
-Related work section is quite extensive.
-In the end I believe that this work is a good contribution to the community.

**Weaknesses:**

-One can hear in the more strenuous experiments that the audio samples have some kind of weird pitch or pace of speaking.
-Paper might not be a very good fit in this venue. Although it has to do with learning representations, NaturalSpeech2 is more fit for a Speech venue such as InterSpeech or ICASSP.

**Questions:**

-Why did the authors not include any experiments with single speaker data like LJSpeech.
-It would be interesting to hear some audio samples with people that have an accent. This has not been explored in the community.
-As an ablation what would be the shortest prompt in seconds that you can give for zero-speech synthesis?
-After the phoneme Encoder you have a Duration and Pitch predictor. Why didn't you also include an Energy Predictor like the authors did in FastSpeech2 since the idea seems to be derived form there?

**Details Of Ethics Concerns:**

They authors address this issue in the conclusions section. After all this is a speech synthesis work and it can be misused in the future. For this venue though I wouldn't want to see this work getting rejected because of this.

---

> ### Author Response · Authors · 2023-11-19
> **Response to Reviewer 69Md**
>
> First of all, we want to thank the reviewer for your careful reading and providing a lot of constructive comments! Below we address the concerns mentioned in the review.
>
> $\textbf{Q1}$: Paper might not be a very good fit in this venue. Although it has to do with learning representations, NaturalSpeech2 is more fit for a Speech venue such as InterSpeech or ICASSP.
>
> $\textbf{A1}$: We believe that our paper is fitting for ICLR.
>
> Firstly, we would like to highlight that NaturalSpeech 2 mitigates the representation issues inherent in large-scale TTS systems. Specifically, previous works utilize multiple discrete token sequences as speech representation, which introduces a dilemma between the codec and the acoustic model (please refer to Sec. 1 for more details). To this end, we employ a single continuous sequence as representation, which helps improve the performance.
>
> Secondly, we employ a new learning paradigm, which involves the use of latent diffusion and continuous representation, to effectively address the issue in the field of speech representation learning in large-scale TTS. This research aligns with the core interests of ICLR, especially in the primary area of 'representation learning for computer vision, audio, language, and other modalities.'
>
> $\textbf{Q2}$: Why did the authors not include any experiments with single-speaker data like LJSpeech?
>
> $\textbf{A2}$: Thank you for your question. We would like to highlight that our work focuses on diverse speech synthesis in terms of speaker identity, prosody, and style, especially the zero-shot text-to-speech synthesis (i.e., on unseen speakers).
>
> For single-speaker datasets such as LJSpeech, firstly, they lack speech diversity since there is only one speaker. Thus, it is not aligned with our focus. Secondly, some previous works, such as [1], have already achieved human-level voice quality. This essentially narrows the scope for further enhancements using this dataset.
>
> [1] Tan, Xu, et al. "Naturalspeech: End-to-end text to speech synthesis with human-level quality."
>
> $\textbf{Q3}$: As an ablation what would be the shortest prompt in seconds that you can give for zero-speech synthesis?
>
> $\textbf{A3}$:  This is a great suggestion to investigate the shortest prompt length. We vary the length of the prompt and report the corresponding SMOS on the LibriSpeech test set. The table below shows that even when reducing the prompt length from 3.0 seconds to 1.0 seconds, NaturalSpeech 2 is still capable of generating high-quality speech. It further demonstrates the robustness and stability of our system despite the reduction in prompt length.
> | Prompt Length| SMOS|
> | ------- | ---- |
> | 3.0 s  |  4.06  |
> | 2.0 s  |  3.98  |
> | 1.0 s  |  3.79  |
>
> $\textbf{Q4}$: Why didn't you also include an Energy Predictor like the authors did in FastSpeech2 since the idea seems to be derived from there?
>
> $\textbf{A4}$: Thank you for your insightful question. The energy predictor is indeed a versatile module. In practice, we have added the energy predictor and did not observe any improvement. For the sake of simplicity in our current model, we let the diffusion model implicitly predict energy information. However, we acknowledge that incorporating an energy predictor could further enhance controllability. We value this suggestion and will take it into consideration for our future research.
>
> $\textbf{Q5}$: Concerns about the potential misuse in the future.
>
> $\textbf{A5}$: Indeed, to protect users from misuse, we will implement a multi-faceted approach, including:
> * Usage Agreement: We will require users to agree not to misuse the technology, which includes not creating synthetic speech without approval.
> * Transparency Measures: We will ensure that any synthetic speech should clearly disclose that the speech is machine-generated to avoid misleading uses.
> * Auditing and Monitoring: There should be regular reviews to detect misuse and taking action against violators.
> * Reporting Mechanism:  We will establish a system for individuals to report any suspected misuse of the technology.

---

> > ### Comment · Reviewer_69Md · 2023-11-19
> >
> > Thank you authors for your details answer. I was in favor of accepting the paper initially, and now I am increasing my score since all of my comments have been addressed.

---

> > > ### Author Response · Authors · 2023-11-20
> > >
> > > Thank you very much for your insightful suggestion and for increasing your score! We greatly appreciate your thorough review and valuable feedback throughout this process.

---

### Official Review · Reviewer_9oT1 · 2023-10-31

**Soundness:** 3 good
**Presentation:** 2 fair
**Contribution:** 3 good
**Rating:** 8
**Confidence:** 5

**Summary:**

This paper describes a TTS model combining a number of modern components these include in-context learning (prompting) a diffusion model to connect conditioning information to latents, and latents defined by an autoencoder for waveform reconstruction.

The resulting model has many of the zero-shot capabilities of LM based TTS that have been presented in recent years, but by maintaining duration prediction for alignment, the model stays robust to a hallucination and dropping errors that impact other generative models.

**Strengths:**

The model contains innovative structures in the in context learning for duration and pitch, and in the diffusion model.  Moreover the overall structuring of these components is novel.

The quality of the model is quite high and provides some important balancing between zero-shot capabilities and robustness compared to alternate models

**Weaknesses:**

The paper is sometimes unclear with regards to what the model components represent and how the components fit together.  For example, the use of SoundStream and wavenet is not obvious.  These are previously published approaches, that are used in novel ways here.  It took multiple readings to understand how they are being used in this paper, and even still i’m not 100% sure that my understanding is correct.  Broadly, the paper relies too heavily on Figure 1.0 to describe how the model fits together.

The argumentation around continuous vs discrete tokens is very hard to follow.  It’s not clear why the discrete token sequence must necessarily be longer than a continuous sequence (Introduction).  The first three pages spend a lot of effort describing why a continuous representation is a better fit for this task.  Then in Section 3.1 “However, for regularization and efficiency purposes we use residual vector quantizers with a very large number of quantizers and codebook tokens to approximate the continuous vectors.  This provides two benefits…” This is a particularly surprising turn of the argument to then go on to describe why discrete tokens are useful here.

The diffusion formulation is too compact to be clearly followed.  Page 5. The following sentence includes a number of ambiguities.  “Then we calculate the L2 distance between the residual vector with each codebook embedding in quantizer j and get a probability distribution with a softmax function, and then calculate the cross-entropy loss between the ID of the ground-truth quantized embedding ej and this probability distribution. Lce−rvq is the mean of the cross-entropy loss in all R residual quantizers, and λce−rvq is set to 0.1”  I’d recommend including an appendix entry or describing each clause separately in place.

**Questions:**

Introduction “the zero-shot capability that is important to achieve diverse speech synthesis” – why is zero-shot necessary for diverse speech synthesis?  Also, for what contexts, and use-cases is diverse speech synthesis necessary?

In the introduction – the checklist between NaturalSpeech 2 and “previous systems” is somewhat strange.  Certainly there are previous systems that are non-autoregressive, or use discrete tokens.  I understand that this is not “all previous systems” but those listed. But why compare only to those three related systems? The introduction and related work draw contrast with a variety of alternate TTS models.

Why use a diffusion model instead of any other NAR model?

When presenting the “prior model” in section 3.2 is the phone encoder, duration predictor and pitch predictor pre-trained to some other target? or is there some other notion of a prior model here?

What is the units used in the L_pitch loss? Hz? log Hz? something else?

The variable z is used in a number of different ways, could this be clarified (e.g. in Figure 2 between the prompt, input to diffusion model and output?)

Section 4.1
Page 6 “Both speakers in the two datasets” are there only 2 speakers in the data sets?
Page 6 what is value of sigma in the sigma-second audio segment as a prompt?

How much loss is incurred by filtering the output by a speech scoring model?  E.g.  are 99% of utterances accepted? or 1%?

Note: VCTK utterances are particularly noisy making is a poor comparison for CMOS, but the comparison to Librispeech is more representative.

Section 4.2 “We apply the alignment tool” – which alignment tool?

What is the variance of the prosodic measures – it’s hard to track whether the differences in Table 3 are significant or not.

“When we disable the speech prompt in diffusion, the model cannot converge” – this seems remarkable.  Why does the model require a speech prompt to learn?

Broader Impacts: What would such a protocol to protect users from misuse of this model look like? Presumably this model can generalize to unseen speakers already – so what protections are in place regarding the use of this model as of publication?

**Details Of Ethics Concerns:**

Zero-shot synthesizers have a strong potential for misuse.

---

> ### Author Response · Authors · 2023-11-19
> **Response to Reviewer 9oT1 [1/3]**
>
> First of all, we want to thank the reviewer for your careful reading and providing a lot of constructive comments! Below we address the concerns mentioned in the review.
>
> $\textbf{Q1}$: The paper is sometimes unclear with regard to what the model components represent and how the components fit together.
> $\textbf{A1}$: Thanks for your careful reading. We have updated the paper and highlighted the updates as follows.
>
> For neural audio codec, firstly, we adopt the SoundStream architecture because: 1) it provides good reconstruction quality and low bitrate, 2) it can convert waveforms into continuous vectors while retaining fine-grained details, and 3) based on the quantized token IDs, we can design the regularization loss, which further enhances the prediction precision. Secondly, we adapt the SoundStream by 1) we use the post-quantized latent $z$, i.e., the output from the RVQ, as the representation for waveform $x$, and as the training target for the diffusion model, 2) we develop the regularization loss $L_{ce-rvq}$.
>
> We adopt WaveNet as the architecture of the diffusion model. Firstly, in our preliminary experiments, we compare various architectures including the vanilla transformer, conformer, UNet and WaveNet. Among these, WaveNet exhibits superior performance in terms of audio quality (CMOS), hence our choice for the diffusion model's architecture. Secondly, to facilitate in-context learning, we incorporate a Q-K-V attention and a FiLM layer in the block. Please refer to Appendix B for more details.
>
> Regarding the interrelation between the components, our system is composed of three components: a prior model, a diffusion model, and an audio codec.  Similar to [1], the prior model (a phoneme encoder and a duration/pitch predictor) encodes the text $y$ into a prior $c$. Then, the diffusion model predicts the speech representation $z$ using prior $c$ as a condition. Finally, the speech representation $z$ is input into the decoder of the audio codec to generate the speech signal $x$.
>
> [1]Ren, Yi, et al. Fastspeech 2: Fast and high-quality end-to-end text to speech.
>
> $\textbf{Q2}$: As shown in Introduction, it’s not clear why the discrete token sequence must necessarily be longer than a continuous sequence.
>
> $\textbf{A2}$: Firstly, we would like to clarify that our paper doesn't assert that the discrete token sequences must necessarily be longer than a continuous sequence. What we claim is that previous works often use multiple RVQ discrete token sequences for speech representation. When these sequences are flattened, it results in a multiple-fold increase in length.
>
> Secondly, it's true that some attempts have been made to address this issue, but they still have their own problems.
> AudioLM[1] predicts the first two RVQ token sequences in the first stage, and predicts the remaining RVQ token sequences in the second stage. However, this approach has two drawbacks: 1) it requires two separate AM models and a two-stage generation process; 2) it still needs the flattening of the code sequences, which also extends the sequence length. VALL-E [2] first predicts the first RVQ token sequence in an AR manner, and then predicts the other RVQ token sequences in a NAR manner, which makes the modeling much more complex compared with one-stage generation. Moreover, MusicGen[3] claims that such parallel modeling may impact system performance.
>
> In contrast, NaturalSpeech 2 uses a single continuous latent instead of multiple discrete tokens for each speech frame. It can prevent the increase in sequence length and support one-stage generation, which helps enhance overall performance.
>
> [1] Borsos, Zalán, et al. Audiolm: a language modeling approach to audio generation.
>
> [2] Wang Chengyi, et al. Neural codec language models are zero-shot text to speech synthesizers.
>
> [3] Copet, Jade, et al. Simple and Controllable Music Generation.

---

> ### Author Response · Authors · 2023-11-19
> **Response to Reviewer 9oT1 [2/3]**
>
> $\textbf{Q3}$: The motivation of adopting RVQ for continuous representation.
>
> $\textbf{A3}$: Thanks for the question.  Firstly, we would like to highlight that we claim predicting the continuous representation instead of the discrete tokens is beneficial to the high-quality TTS system, while the quantization of audio codec can be regarded as a regularization for the continuous space.
>
> Secondly, theoretically, the use of an infinite number of quantizers can approximate a continuous space. In our paper, we adopt more codebooks than [1,2] to achieve a better approximation (please refer to Appendix C).
>
> Finally, the quantization audio codec can bring the following benefits. 1) Data storage efficiency. By using RVQ codec, we do not need to directly store the latent representations, but store the code ID which can recover the latent by looking up the codebook during training. For example, instead of storing 256-dimensional FP32 (256 x 32 bits) latent representations directly, we only need to store 16-dimensional INT16 quantized IDs (16 x 16 bits), resulting in a $32\times$ increase in storage efficiency. 2) Based on the quantized IDs, we designed a regularization loss, $L_{ce-rvq}$, which further enhances the prediction precision.
>
> [1]Borsos Zalán, et al. Audiolm: a language modeling approach to audio generation.
>
> [2]Wang Chengyi, et al. Neural codec language models are zero-shot text to speech synthesizers.
>
> $\textbf{Q4}$: The description of CE-RVQ loss is not clear.
>
> $\textbf{A4}$: Thanks for your advice. We have provided more details in Appendix C.3 in the revised version. We also provide the detailed formulation of CE-RVQ loss as follows:
>
> Moreover, to clarify, for each residual quantizer $j \in [1, R]$, we first get the residual vector:
> \begin{equation}
> z_j = z_0 - \sum^{j-1}_{m=1}\hat{e}^{m},
> \end{equation}
>
> where $\hat{e}^m$ is the ground-truth quantized embedding in the $m$-th residual quantizer. Then we calculate the L2 distance between the residual vector with each codebook embedding in quantizer $j$ and get a probability distribution as follows:
> \begin{equation}
> l_i = ||z_j - e^j_i||_{2},
> \end{equation}
>
> \begin{equation}
> s_i = \frac{e^{-l_i}}{\sum^{N_j}_{k=1}{e^{-l_k}}},
> \end{equation}
>
> where $N_j$ is the code number of residual quantizer $j$, and $s_i$ is the probability of code $i$ in codebook $j$. Finally, we can calculate the cross-entropy loss of residual quantizer $j$ given the ground-truth code index which is denoted as $L_{ce, j}$. The final CE-RVQ loss is shown as follows: \begin{equation}
> L_{ce-rvq} = \sum_{j=1}^{R}{L_{ce, j}}
> \end{equation}
>
> $\textbf{Q5}$: Why is zero-shot necessary for diverse speech synthesis?  For what contexts, and use-cases is diverse speech synthesis necessary?
>
> $\textbf{A5}$:  Thanks for your careful reading.
>
> Firstly, diverse speech synthesis aims to generate natural and human-like speech with rich diversity in speaker identity (e.g., gender, accent, timbre), prosody, style, et.al. However, it is difficult and expensive to collect all kinds of speeches to capture such diversity. And for zero-shot synthesis, it is valuable for generating speech with a variety of elements like timbre, prosody, and emotion, without the need for explicit training for these attributes, which significantly eases the difficulty of data collection and broadens the potential diversity in speech synthesis.
>
> Secondly, diverse speech synthesis is necessary in various contexts and use-cases.  For instance,
> 1) Voice Assistants: To make interactions more personalized and engaging, voice assistants need to be able to generate a variety of voices.
>
> 2) Entertainment: In industries like animation or gaming, diverse speech synthesis can be used to create a wide array of unique character voices.
>
> 3) Language Learning: Diverse speech synthesis can help language learners by providing a variety of accents and pronunciations to learn from.
>
> $\textbf{Q6}$: The checklist between NaturalSpeech 2 and “previous system” is not clear. This is not “all previous systems” are included.
>
> $\textbf{A6}$:  We would like to clarify that, as highlighted in the caption, the comparison in the checklist is specifically made with previous $\textit{large-scale TTS systems}$, rather than all models.  We have noted the potential for misunderstanding and have revised the checklist accordingly.

---

> ### Author Response · Authors · 2023-11-19
> **Response to Reviewer 9oT1 [3/3]**
>
> $\textbf{Q7}$: Why use a diffusion model instead of any other NAR model?
>
> $\textbf{A7}$: Compared to NAR methods, the diffusion model stands out as it incorporates multiple iterations in inference. This allows for a progressive refinement over multiple iterations, which helps enhance the modeling capability, and makes it particularly effective high-quality generation.
>
> To demonstrate this effectiveness, we employ FastSpeech2[1] as a NAR baseline in our paper (in Sec. 4.2). We adapt it by adding cross-attention on speech prompts for zero-shot synthesis. Furthermore, we also change the prediction target from the mel-spectrogram to the latent representation. For a fair comparison, we scale it to 400M parameters and trained it on the same MLS dataset. As shown in Table 2, the significant performance gap indicates limitations in its modeling capability.
>
> [1] Ren, Yi, et al. Fastspeech 2: Fast and high-quality end-to-end text to speech.
>
> $\textbf{Q8}$: When presenting the “prior model” in Sec. 3.2 is the phone encoder, duration predictor and pitch predictor pre-trained to some other target? or is there some other notion of a prior model here?
>
> $\textbf{A8}$: We refer to the phoneme encoder and duration/pitch predictor collectively as the "prior model". As mentioned in equation (6), the prior model is trained jointly.
>
> $\textbf{Q9}$: What is the units used in the L_{pitch} loss? Hz? log Hz? something else?
>
> $\textbf{A9}$: Thanks for your careful reading. We normalize the pitch to zero mean and unit variance.
>
> $\textbf{Q10}$: The variable z is used in a number of different ways, could this be clarified (e.g. in Figure 2 between the prompt, input to diffusion model and output?)
>
> $\textbf{A10}$: Thanks for your careful reading. We would like to clarify that the notation 'z' is used throughout the article to consistently represent the latent audio representation encoded by the codec.
>
> $\textbf{Q11}$: Sec. 4.1 Page 6 “Both speakers in the two datasets” are there only 2 speakers in the data sets?
>
> $\textbf{A11}$: We sincerely apologize for the confusion. The phrasing was a typo and should be "All speakers in the two datasets", not "Both speakers in the two datasets."  The LibriSpeech test-clean dataset indeed includes 40 distinct speakers, and the VCTK dataset includes 108. We have corrected this typo in the revised version. Thank you for pointing it out.
>
> $\textbf{Q12}$: Page 6 what is value of sigma in the sigma-second audio segment as a prompt?
>
> $\textbf{A12}$: Thanks for your careful reading. As mentioned in Sec. 4.2 'Generation Similarity', we set $\sigma$ to $3$ in Sec.4.
>
> $\textbf{Q13}$: How much loss is incurred by filtering the output by a speech scoring model?
>
> $\textbf{A13}$: There may be some misunderstanding about the “loss incurred by filtering”. We use a speech scoring model for reranking. We sample 10 candidates and select the best one.
>
> $\textbf{Q14}$: Sec. 4.2 “We apply the alignment tool” – which alignment tool?
>
> $\textbf{A14}$: We use the internal alignment tool, which is similar to Montreal Forced Aligner[1].
>
> [1] McAuliffe, Michael et al. Montreal Forced Aligner: Trainable Text-Speech Alignment Using Kaldi.
>
> $\textbf{Q15}$: What is the variance of the prosodic measures?
>
> $\textbf{A15}$: We appreciate your insightful question. We provide the measures with a 95% confidence interval. For pitch, the measures (Mean, Std, Skew, Kurt) are $10.11_{\pm 0.08}, 6.18_{\pm 0.06}, 0.50_{\pm0.01},1.01_{\pm 0.03}$, respectively. For duration, the measures are $0.65_{\pm 0.01}, 0.70_{\pm 0.01}, 0.60_{\pm 0.01}, 2.99_{\pm 0.07}$, respectively. These results demonstrate that the improvements in Table 3 are statistically significant.
>
> $\textbf{Q16}$: “When we disable the speech prompt in diffusion, the model cannot converge” – this seems remarkable. Why does the model require a speech prompt to learn?
>
> $\textbf{A16}$: In practice, we find that the absence of a speech prompt, which supplies the speaker identity, may cause instability during the training phase. This may be attributed to the massive diversity of speech data given the large number of speakers in the training dataset.
>
> $\textbf{Q17}$: Broader Impacts: What would such a protocol to protect users from misuse of this model look like?
>
> $\textbf{A17}$: Thanks for your question. Indeed, to protect users from misuse, we will implement a multi-faceted approach, including:
>
> * Usage Agreement: We will require users to agree not to misuse the technology, which includes not creating synthetic speech without approval.
>
> * Transparency Measures: We will ensure that any synthetic speech should clearly disclose that the speech is machine-generated to avoid misleading uses.
>
> * Auditing and Monitoring: There should be regular reviews to detect misuse and taking action against violators.
>
> * Reporting Mechanism:  We will establish a system for individuals to report any suspected misuse of the technology.

---

> ### Author Response · Authors · 2023-11-22
>
> We sincerely appreciate your comprehensive feedback. As the discussion period nears its end, we would like to know if there are any additional questions. We are glad to answer them.

---

> > ### Comment · Reviewer_9oT1 · 2023-12-04
> >
> > Thank you for the thorough response to the review.  I appreciate it, and hope you agree that the edits strengthen the paper.  I've updated my overall assessment score.

---

### Official Review · Reviewer_c9Qd · 2023-11-01

**Soundness:** 3 good
**Presentation:** 2 fair
**Contribution:** 3 good
**Rating:** 8
**Confidence:** 3

**Summary:**

This paper presents NaturalSpeech 2, a non-autoregressive TTS model that employs a diffusion mechanism to generate quantized latent vectors from neural audio codecs. It shows enhanced zero-shot TTS performance relative to the state-of-the-art large-scale neural codec language model. The proposed approach exhibits advancements in sample quality, intelligibility, robustness, speaker similarity, and generation speed when benchmarked against the baseline method. The authors further validate the superiority of their method over other alternatives via comprehensive qualitative and quantitative evaluations.

**Strengths:**

* The paper effectively tackles several major challenges inherent to non-autoregressive TTS modeling at scale.
* The authors have carried out robust and wide-ranging experiments, yielding detailed results.
* The reference list is both extensive and comprehensive.

**Weaknesses:**

The proposed method's intricate modeling could hinder its extension to other applications. While the introduced model applies diffusion, it necessitates two additional losses and requires supplementary modules like a pitch predictor, prompt encoder and the second attention block. As an example, the recent state-of-the-art flow-matching based TTS method, VoiceBox [1] consists of rather simple model architecture; the flow-matching based duration predictor and audio model.

[1] Le, Matthew, et al. "Voicebox: Text-guided multilingual universal speech generation at scale." arXiv preprint arXiv:2306.15687 (2023).

**Questions:**

Given the concerns mentioned in the above weaknesses, it would be interesting to see if the proposed method could be adapted or refined to reduce its dependency on additional modules without increasing complexity or compromising sample quality.

---

> ### Author Response · Authors · 2023-11-19
> **Response to Reviewer c9Qd**
>
> First of all, we want to thank the reviewer for your careful reading and providing a lot of constructive comments! Below we address the concerns mentioned in the review.
>
> $\textbf{Q1}$: The proposed method necessitates two additional losses and requires supplementary modules like a pitch predictor, prompt encoder and the second attention block.  It would be interesting to see if the proposed method could be adapted or refined to reduce its dependency on additional modules.
>
> $\textbf{A1}$: Thanks for your careful reading.
> 1) For the prompt encoder, it is adopted to encode the prompt speech for high-quality zero-shot text-to-speech synthesis, which is also widely used in previous works, such as [1,2,3].
>
> 2) For the second attention block, it is crucial to NaturalSpeech 2 for high-quality synthesis. Intuitively, the prompt needs to provide the model with aspects like timbre, prosody, energy, background, et.al. Thus, we adopt $m$ query tokens in the second attention block to model these important aspects. With experiments in Sec. 4.3 (Ablation Study’s w/o. query attn), we can find this component is beneficial to the audio generation quality.
>
> 3) For the CE-RVQ loss, it can significantly enhance the performance while incurring low computational cost. Intuitively, it is a regularization constraint inspired by the hierarchical structure of the audio codec. To ensure high-quality generation, we constrain the latent distribution and propose the CE Loss by matching the predicted $z_0$ with each quantizer. With experiments in Sec. 4.3 (Ablation Study’s w/o. CE loss), we can find that both the CMOS and WER will degrade by removing it, which demonstrates the effectiveness of CE-RVQ loss.
>
> 4) For the pitch predictor, it is a flexible module in NaturalSpeech 2. We have conducted the ablation experiment by removing the pitch predictor and observed no noticeable decline in performance. However, it significantly enhances the controllability of pitch, which is necessary for singing voice synthesis (please refer to Sec. 4.4 for more information). Thus, we still adopt the pitch predictor in NaturalSpeech 2.
>
> [1] Arik, Sercan, et al. Neural voice cloning with a few samples.
>
> [2] Jia, Ye, et al. Transfer learning from speaker verification to multispeaker text-to-speech synthesis.
>
> [3] Jiang, Ziyue, et al. Mega-TTS: Zero-Shot Text-to-Speech at Scale with Intrinsic Inductive Bias.

---

> > ### Comment · Reviewer_c9Qd · 2023-11-22
> > **Response to Authors**
> >
> > I am grateful for the detailed response from the authors. They have provided rationales explaining the necessity of their proposed method's modeling. While I understand from their explanation and experimental results that the application of various modules significantly enhances performance, it appears to lean more towards engineering efforts. To strengthen the scientific contribution of this research, it could have been further validated by, for example, adding experimental results and analyses to demonstrate the effectiveness of these modules in applications beyond Text-to-Speech (TTS), such as in singing voice synthesis and various other tasks.

---

> > > ### Author Response · Authors · 2023-11-22
> > >
> > > Thanks for your insightful response!
> > >
> > > As suggested, we have done similar ablation experiments before in singing voice synthesis, and thus, we provide the following experimental results as supplementary information. 1) For CE loss, we find that removing the CE loss can largely worsen the singing generation quality, which is similar to the results in TTS. It further demonstrates the effectiveness of CE loss. 2) For pitch predictor, we find that without modeling pitch explicitly, firstly, the model loses the ability to generate singing audio following the specific given music scores. Secondly, the singing audio has bad quality.
> > > We think singing voice synthesis is much harder and the pitch information can be regarded as a strong control signal to guide the generation process.
> > >
> > > Finally, we appreciate your feedback on extending to various other tasks and exploring a more powerful and simple architecture. We would like to leave it for future work.

---

### Comment · Area_Chair_ztX5 · 2023-11-21
**Reminder to reviewers to participate in the author/reviewer discussion**

Dear reviewers, this is a reminder that the author/reviewer discussion period ends November 22.

This discussion is indeed supposed to be a dialog, so please respond to
the comments from the authors.

AC

---

### Meta-Review · Area_Chair_ztX5 · 2023-12-05

**Metareview:**

## Scientific claims and findings
The paper proposes a model architecture for large-scale, multi-speaker, in-the-wild text-to-speech conversion. The model comprises (1) a neural audio codec that can map audio waveforms to discrete vectors and back; (2) a diffusion model that generates sequences of latent discrete vectors given conditioning information; and (3) a prior model comprising a phoneme encoder, duration predictor, and pitch predictor that provide the conditioning information to the diffusion model. In contrast to previous large-scale TTS models, the discretizaton of the latent vector in the audio codec is done at a very high resolution, which improves audio quality. Use of the diffusion model as a generative mechanism avoids the challenge faced by models that use a generative language model, namely that improving output quality by using a higher resolution discretization entails either modeling much longer sequences or implementing a complex decoding mechanism. To solve the one-to-many problem inherent in TTS (one phoneme sequence can generate countless different acoustic realizations), the prior and diffusion models use a speech prompting mechanism. Experiments on Librispeech and VCTK using both objective and subjective metrics show that the proposed model outperforms competing large-scale TTS models, and ablation studies show the importance of various components of the model, including the speech prompting, cross-entropy loss on the discrete latent codes, and the attention on the speech pprompt using a randomly initialized query sequence to prevent leakage of details.

## Strengths
- Very strong set of experiments to evaluate the model against the state of the art, and good ablations showing the importance of different components of the model.
- Interesting combination of a diffusion model and discrete neural audio codec for speech synthesis.
- Explicit modeling of duration and pitch in the prior model allows for controlling these parameters during synthesis, which may be important in some TTS applications.
- Very good coverage and discussion of related work.

## Weaknesses
- The discussion of continuous vs. discrete vectors in the codec and in the synthesis process, while improved over the first draft, is still somewhat confusing. The initial discussion in the paper, basically everything mentioning continuous vectors up until the last paragraph of Section 3.1 (including the Section title for 3.1!), leads the reader to expect that the model uses continuous vectors, but in practice discrete vectors are actually used. This comes across as somewhat deceptive. I think it is better to frame the discussion as saying that, in principle, using continuous vectors would completely solve the problem of getting good generation quality without blowing up sequence lengths or resorting to multi-step decoding strategies, but that there are practical reasons to use discrete representations, so the solution is to use relatively high-resolution discrete representations, instead of even claiming that the model uses continuous representations.

**Justification For Why Not Higher Score:**

- While I agree with the authors that the paper is definitely in scope for ICLR, there is a lot of very TTS-specific content to this paper that will have less of an audience at ICLR than some other papers, so I think it is better as a spotlight than as an oral.
- The discussion around continuous vs. discrete representations still needs to be clarified (mentioned above under "Weaknesses").

**Justification For Why Not Lower Score:**

- This is a high-quality paper with all four reviewers recommending acceptance unambiguously.
- The core idea of using a diffusion model to generate latent codes for generation is an interesting one that could be applied more broadly.

---

### Decision · Program_Chairs · 2024-01-16

Accept (spotlight)